# Towards Transparent Time Series Forecasting

**Krzysztof Kacprzyk**
University of Cambridge
kk751@cam.ac.uk

**Tennison Liu**
University of Cambridge
tl522@cam.ac.uk

**Mihaela van der Schaar**
University of Cambridge
The Alan Turing Institute
mv472@cam.ac.uk

## Abstract

Transparent machine learning (ML) models are essential for ensuring interpretability and trustworthiness in decision-making systems, particularly in high-stakes domains such as healthcare, finance, and criminal justice. While transparent machine learning models have been proposed for classification and regression, time series forecasting presents some unique challenges for ensuring transparency. In particular, currently used bottom-up approaches that focus on the values of the time series at specific time points (usually regularly spaced) do not provide a holistic understanding of the entire time series. This limits the applicability of ML in many critical areas. To open up these domains for ML, we propose a top-down framework of bi-level transparency, which involves understanding the higher-level trends and the lower-level properties of the predicted time series. Applying this framework, we develop TIMEVIEW, a transparent ML model for time series forecasting based on static features, complemented with an interactive visualization tool. Through a series of experiments, we demonstrate the efficacy and interpretability of our approach, paving the way for more transparent and reliable applications of ML in various domains.

## 1 Introduction

**Why do we need transparent models?** eXplainable Artificial Intelligence (XAI) methods are broadly divided into transparent models and post-hoc explanation techniques (Barredo Arrieta et al., 2020). Transparent (also called glass box) models are crucial in many settings involving high-stakes decisions such as healthcare or credit scoring (Rudin, 2019). As these models are interpretable by design, they are by themselves understandable. Such understanding, apart from being mandated by certain regulatory bodies (Goodman & Flaxman, 2017), is needed, for instance, to improve robustness, detect biases, evoke trust, or certify model compliance with legislation (Barredo Arrieta et al., 2020). Many transparent machine learning models that issue *static predictions* have been proposed. That includes Linear/Logistic Regression, Generalized Additive Models (Hastie & Tibshirani, 1986), and Decision Trees (Hu et al., 2019). By definition, these methods are not directly applicable to time series forecasting—when we want to predict a whole trajectory rather than a single label. Although they can be adapted, they exhibit poor performance (see Section 7).

**Challenges of time series forecasting: limitations of a bottom-up approach.** In contrast to a single-label output (as in classification or regression), understanding the change in the trajectory is more complicated as it is an entire function (described by many values). As time series forecasting remains an under-studied field of XAI (Barredo Arrieta et al., 2020), current techniques usually resolve to a *bottom-up* approach. This means they focus on the values of the trajectory at individual time points (usually regularly spaced). For instance, the importance scores in saliency methods are calculated for different prediction horizons (Leung et al., 2023). This may be sufficient when we are interested in a particular time point (e.g., 5-year survival rate), but we often want to comprehend the whole trajectory at once. For instance, when administering a drug, we may be less interested in the concentration of the drug every few hours but rather in understanding the entire curve, including properties like the peak plasma concentration and the time when it is achieved (Han et al., 2018).

**Bi-level transparency for time series forecasting: a top-down approach.** We propose a *top-down* approach to trajectory comprehension and consequently two levels of transparency for time series forecasting: (level 1) understanding how the *trend* (the general shape of the trajectory) changes as we

modify the input, and (level 2) understanding how the *properties* of the current trend (e.g., minimum value) change as we modify the input. To illustrate this, let us consider the following example.

> ### Forecasting a tumor volume trajectory from patient's baseline covariates and drug dose
>
> Understanding a model like this may include answering questions such as: (Liao et al., 2020)
>
> - **What if**: "What would happen to the model's prediction if a specific covariate changes?"
> - **How to be that**: "How should the covariates be modified to get a different prediction?"
> - **How to still be this**: "What range of drug dose values keeps the prediction the same?"
>
> We characterize the difference between predicted trajectories on two levels, which enables us to answer concrete questions about each level such as
>
> | Level 1 (trends) | Level 2 (properties) |
> |---|---|
> | "Would the predicted tumor volume keep *decreasing* if we adjusted the treatment?" | "What feature changes would lower the *minimum* tumor volume?" |

We explain trends and properties in detail in Section 2 and then formalize them in Section 4. It is worth noting that answering such questions with the current bottom-up approaches may often be futile since the notion of a "different prediction" (based on individual time points or norms such as $L^p$) may be non-interpretable (see Section 2) or simplistic. We demonstrate how our framework can enable answering such questions in Figure 1 and more thoroughly in Appendix E.1.

**Time series forecasting based on static features.** Our work focuses on understanding the change in the predicted trajectory. However, time series models take many types of inputs, including time series and exogenous features. To provide a clear exposition of our framework, we focus on one specific input type: static features. Time series forecasting based on static features has applications in many domains ranging from finance through medicine and pharmacology to physics (see Section 3). In Section 5, we introduce TIMEVIEW—a transparent ML model for time series forecasting based on static features. As with many transparent models (e.g., GAMs, decision trees), model visualization is crucial for its interpretability. In Section 7, we demonstrate a visualization tool based on interactive plots that allows for the examination of both the higher and lower-level features of the predicted trajectories and how they change based on the input (Figure 1).

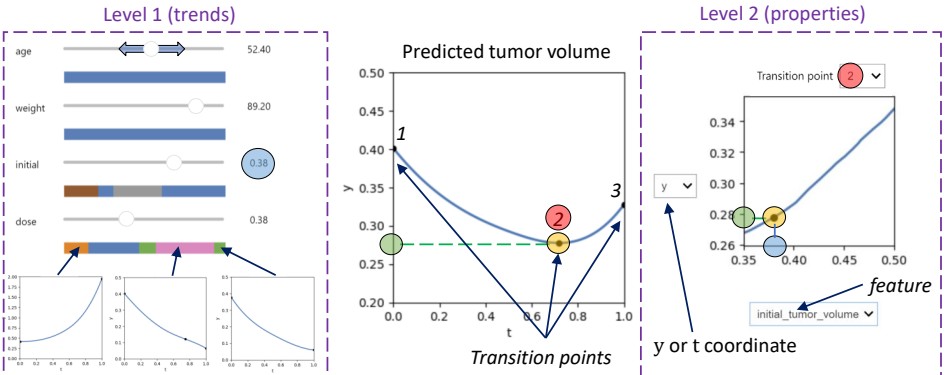

Figure 1: Snapshot of our dynamical visualization of TIMEVIEW. Our model adheres to *bi-level transparency*—a *top-down* approach that focuses on the trend of the trajectory (Level 1) and the properties of the particular trend, e.g., transition points (Level 2). The left panel shows how the trajectory trend changes when one of the features is perturbed. For instance, the tumor will increase if we lower the dose and decay if we increase the dose. The right panel investigates the position ($y$-coordinate) of the second transition point (local minimum) as the initial tumor volume changes.

**Contributions.** We introduce *bi-level transparency*, a novel top-down framework for time series forecasting that allows for a holistic understanding of the entire trajectory through trends and properties (Section 2). We formalize it by introducing the notions of *motifs* and *compositions* (Section 4). Based on the new formalism, we develop TIMEVIEW, Time series Interpretable Model with Effective VIsualization (Section 5). We demonstrate how its visualization aids in model comprehension while exhibiting only a minor performance drop compared to black-box models (Section 7).

## 2 TRANSPARENCY FOR TIME SERIES FORECASTING

### 2.1 SETUP

**Time series forecasting.** A general ML model is a function $f$ mapping samples from the input space $\mathcal{X}$ to the output space $\mathcal{Y}$. We say that $f$ issues *static predictions* when $\mathcal{Y}$ is a subset of $\mathbb{R}$ (a *regression model*) or a finite set of labels $\{1, \dots, K\}$ (a *classification* model).[1] In contrast, we define $f$ to be a *time series forecasting* model (or just *forecasting model*) when $\mathcal{Y}$ is a space of *trajectories*. A trajectory is a function $y : \mathcal{T} \to \mathbb{R}$, where $\mathcal{T} \subset \mathbb{R}$ is a set of time points. Although the conceptual framework in Section 2 is agnostic to the nature of $\mathcal{T}$, our work focuses on settings where $\mathcal{T}$ is an interval $[0, T] \subset \mathbb{R}$, where $T \in \mathbb{R}$ is a *time horizon*, and the underlying trajectory is continuous. Note, in practice, we only observe discrete samples of $y$, which may be irregular and noisy.

**Transparency.** We assume the following general definition of transparency: *ML model is transparent if we can understand the impact of the inputs on the prediction*. In particular, how changing one (or a few) of the features would impact the output. This is crucial for counterfactual reasoning (e.g., "What would the model predict if the patient was female?") or detecting anomalies (e.g., "Why does the model assign a significantly higher risk score if the patient's age is changed from 64 to 65?").

**Comprehending the change in the output.** As discussed in Section 1, understanding the change in the output is crucial for answering important questions about the model (e.g., "What would happen to the model's prediction if a specific feature changes?"). In a static setting, when the prediction is a single label, understanding the change in the output is relatively straightforward as only a few things can happen. In regression, the target variable can decrease, increase, or remain constant. In classification, the target variable can change from one option to another among a finite number of classes. In time series forecasting, when the prediction is a trajectory, understanding the change in the output is challenging because there are numerous ways a function can change (discussed further in Appendix E). Moreover, these changes need to be interpretable for humans.

### 2.2 BOTTOM-UP: CURRENT XAI APPROACH TO TRAJECTORY COMPREHENSION

As trajectory is a function $y : \mathcal{T} \to \mathbb{R}$, current XAI techniques for time series forecasting focus on understanding the impact of the inputs on $y(t)$ for a particular $t \in \mathcal{T}$. For instance, the values in saliency methods are calculated independently for different prediction horizons (Leung et al., 2023) (and might be later aggregated). Inspired by the motivations of rough path theory (Lyons, 2014; Fermanian et al., 2023), we call the current comprehending strategy *bottom-up*. It means the trajectory is understood by looking at its values at individual time points, and subsequently, more information is gained by looking at more points[2]. However, we argue that this strategy for trajectory understanding is *not* optimal in many scenarios. In particular, it is not a natural way for people to understand trajectories, and it is challenging to convey time-varying trends and global features by simply looking at individual time points in isolation.

**Inconsistent with the natural way people understand trajectories.** Standard representation of a trajectory $y : [0, T] \to \mathbb{R}$ is a line graph. Research on graph comprehension (Zacks & Tversky, 1999; Xi, 2010) suggests that people understand line graphs in terms of trends rather than individual values. For instance, "when $x$ increases, $y$ also increases". They also tend to focus on the minimum and maximum values and trend reversals (Carswell et al., 1993). Thus, understanding a (continuous) trajectory by individual values is unnatural for humans. See Appendix E for more details.

**Increased cognitive load.** As mentioned above, the bottom-up approach requires an increasing number of values to understand the trajectory better. This becomes problematic when we want to understand any change in the trajectory, as it places the cognitive burden on the human interpreter to piece together changes in trends from changes at individual time steps.

**Unsuitable for global features.** A bottom-up approach may be sufficient when we are interested in a particular time point (e.g., 5-year survival rate) or when there are only a few time points of interest. However, we often want to comprehend the whole trajectory at once. For instance, when administering a drug, we are interested in understanding the entire drug concentration *curve*, including properties like peak plasma concentration and the time when it is achieved (Han et al., 2018).

---

[1]Another example of a model issuing static predictions is a *multi-output regression* where $\mathcal{Y} \subset \mathbb{R}^K$.

[2]Note, "bottom-up" refers to how the trajectory is comprehended, not how the prediction is generated

## 2.3 TOP-DOWN: NEW APPROACH TO TRAJECTORY COMPREHENSION

To address the shortcomings of the bottom-up approach, we propose a *top-down* approach to understanding a trajectory. It is motivated by the fact that humans tend to describe trajectories by referring to the *trends* and *properties* it exhibits rather than just the values it attains (Carswell et al., 1993). Consider the natural language descriptions of trajectories presented in Table 1. In all these examples, we have a *trend*—the general shape of the function (e.g., "increasing", "stays below"), and *properties*—the details of the particular trend (e.g., "for the last 10 years", "below 100mg/dl").

Table 1: When we describe a trajectory, we often refer to the trends and properties it exhibits. We use this observation in our definition of *bi-level transparency*, which ultimately informs the design of our method. The table shows three examples of descriptions of trajectories with their corresponding trends and properties.

| Description | Trend | Properties | Visualization |
|---|---|---|---|
| "The GDP has been steadily increasing for the last 10 years" | increasing | for the last 10 years | |
| "The blood sugar level in non-diabetic patients should stay below 100mg/dl while fasting" | stay below | below 100mg/dl | |
| "Tumor volume decreases, obtains a minimum after 6 months, and then increases" | decreases then increases | minimum at 6 months | |

The top-down approach addresses shortcomings of the bottom-up approach, i.e., it is more consistent with the natural way people understand trajectories and conveys time-varying trends and global features in an interpretable way. Moreover, it is also compatible with the scientific approach to analyzing various trajectories. For instance, while studying dynamical systems, we are often interested in understanding bifurcations—a qualitative change in the behavior of a system as the parameter changes (Blanchard et al., 2012). This corresponds to understanding the inputs where the trend of the trajectory changes.

**Bi-level transparency: understanding how the trends and properties change.** By using the top-down approach above, we do not need all the trajectory values to understand it. Instead, we can focus on the trends and properties of the trajectory and only access the exact values when necessary. This is how we can achieve an interpretable model: instead of tracking the individual values of the trajectory (as in bottom-up approaches), we track how the trends and properties of the trajectory change as we vary the input. Thus, we refine the definition of transparency and adapt it specifically for time series forecasting. We call it *bi-level transparency*.

A time series forecasting model is (bi-level) transparent if the following holds.

- (Level 1) We can understand the impact of the input on the *trends* of the trajectory.
- (Level 2) We can understand the impact of the input on the *properties* of a given trend.

## 3 TIME SERIES FORECASTING FROM STATIC FEATURES

Bi-level transparency *unweaves* the "output" part of transparency into two separate objects: trends and properties. Thus, it provides a concrete answer to the question: what does it mean to understand the change of the *output*? However, time series models may take many types of inputs, including static features, information about the future (e.g., upcoming holiday dates), and other exogenous time series (Lim et al., 2021). To provide a clear exposition of our framework, develop formalism, and demonstrate a practical implementation, we focus on settings where inputs are static features.

**Real life settings.** Time series forecasting from static features is frequently encountered in medicine and pharmacology, where we are interested in predicting the disease progression or the drug concentration based on the patient's covariates. Static features can also include the dosage/strength of the treatment or even the time and type of intervention. If necessary, one or a few initial observations at pre-specified times can also be considered to be static features. More examples of such scenarios can be found in finance (predicting stock values from the company's static data), time-to-event problems (predicting the survival or the hazard function), or modeling any 1D dynamical system from

its initial conditions. In some scientific or engineering domains, time can be even replaced by other continuous variables. For instance, when modeling stress-strain or current-voltage curves.

**Problem formulation.** Let $T \in \mathbb{R}$ be the *time horizon*. Each sample consists of static features $\boldsymbol{x}^{(d)} \in \mathbb{R}^M$, where $M \in \mathbb{N}$ is the number of features, and a discretely sampled trajectory $\boldsymbol{y}^{(d)} \in \mathbb{R}^{N_d}$ at times $\boldsymbol{t}^{(d)} \in \mathbb{R}^{N_d}$, where $N_d \in \mathbb{N}$ is the number of measurements for the $d^{\text{th}}$ sample. We assume that $\boldsymbol{y}^{(d)}$ consists of noisy samples of some true underlying continuous trajectory $y_*^{(d)} : [0, T] \to \mathbb{R}$. Given a dataset $\{\boldsymbol{x}^{(d)}, \boldsymbol{y}^{(d)}, \boldsymbol{t}^{(d)}\}_{d=1}^D$, the task is to find a model that matches static covariates $\boldsymbol{x} \in \mathbb{R}^M$ to a trajectory $\hat{y} : [0, T] \to \mathbb{R}$ such that $\hat{y}$ minimizes the expected value of $\frac{1}{T} \int_0^T (\hat{y}(t) - y_*(t))^2 dt$ for all test samples. We denote the class of predicted trajectories as $\hat{\mathcal{Y}}$.

## 4 MOTIFS AND COMPOSITIONS

In this section, we propose a way to formalize the notion of a trend by defining the *composition* of a trajectory. The composition is a sequence of *motifs* where each motif describes the current "shape" of the trajectory at a specific interval. For instance, we can choose a set of three motifs: "increasing" ($i$), "decreasing" ($d$), and "constant" ($c$). Then, we can divide a trajectory into a few segments, so each can be classified as being in one of these motifs throughout the interval. Thus, we can assign a sequence of motifs to this trajectory - a composition. For instance, a ReLU function on [-1,1] has a composition ("constant", "increasing") or just $(c, i)$, whereas a *sin* on the interval $[0, 2\pi]$ has a composition $(i, d, i)$. The motifs can be chosen based on the application and the required granularity. The points between motifs are called *transition points*, and their coordinates can be mapped to the properties of a trend (see Figure 2)

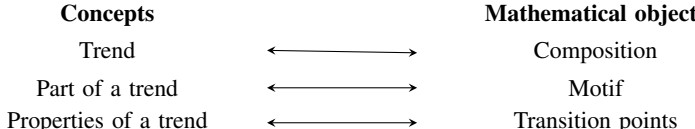

| Concepts | | Mathematical objects |
|---|---|---|
| Trend | $\longleftrightarrow$ | Composition |
| Part of a trend | $\longleftrightarrow$ | Motif |
| Properties of a trend | $\longleftrightarrow$ | Transition points |

Figure 2: Correspondence between concepts in Section 2 and mathematical objects in Section 4

**Notation.** We say $I$ is an interval (of $\mathbb{R}$) if it is an open interval, closed interval, or half-closed interval. The interval has to contain more than one point. We denote the set of all intervals on $\mathbb{R}$ as $\mathcal{I}$. Let $c \in \mathbb{R}$, we denote the shifted interval as $I + c = \{x + c \mid x \in I\}$. Let $I \subset \mathbb{R}$ be any interval, we call any function $f : I \to \mathbb{R}$ an *interval function* and we denote its domain as $\text{dom}(f)$. We denote the set of all interval functions as $\mathcal{F}$.

**Definition 1** (Motif). A motif $s$ is a binary relation between the set of interval functions $\mathcal{F}$ and the set of intervals $\mathcal{I}$ (i.e., $s \subset \mathcal{F} \times \mathcal{I} = \{(f, I) \mid f \in \mathcal{F}, I \in \mathcal{I}\}$). We denote $(f, I) \in s$ as $f|I \sim s$ and read it as "$f$ on $I$ has a motif $s$". Each motif $s$ needs to be:

- *well-defined*, i.e., for any $f \in \mathcal{F}$, and any $I \in \mathcal{I}$,
$$f|I \sim s \implies I \subseteq \text{dom}(f) \tag{1}$$
- *translation-invariant*, i.e., for any $I \in \mathcal{I}$, and any $f \in \mathcal{F}$,
$$f|I \sim s \iff f \circ (x - c)|(I + c) \sim s \; \forall c \in \mathbb{R} \tag{2}$$

Now, we would like to assign a minimal sequence of motifs to a given trajectory: a *composition*.

**Definition 2** (Composition). Let $f : I \to \mathbb{R}$ be an interval function and $\mathcal{S}$ be a set of motifs. A *motif sequence* of $f$ in $\mathcal{S}$ is a finite sequence of motifs $(s_1, \ldots, s_d)$, such that there exists an interval partition[3] $(I_1, \ldots, I_d)$ of $I$ such that $f|I_j \sim s_j \; \forall j \in [d]$. A *composition* of $f$ in $\mathcal{S}$ is the shortest *motif sequence* of $f$ in $\mathcal{S}$. The points between the intervals are called the *transition points*. The set of all compositions for a given set of motifs $\mathcal{S}$ is denoted by $\mathcal{C}_{\mathcal{S}}$. A set of motifs $\mathcal{S}$ is called *compatible* with a subset $\mathcal{F}' \subset \mathcal{F}$ if for every $f \in \mathcal{F}'$ there exists a unique composition, denoted $\mathcal{C}_{\mathcal{S}}[f]$

Compatibility between the set of motifs and the set of trajectories is crucial for an ML model that employs bi-level transparency as we want to assign a composition to every possible prediction unambiguously and, in turn, to every feature vector. We call this assignment a *composition map*.

---

[3]For a definition of interval partition, see Appendix A

**Definition 3** (Composition map). Let a set of motifs $\mathcal{S}$ be compatible with some subset $\mathcal{F}' \subset \mathcal{F}$. Let $g : \mathbb{R}^M \to \mathcal{F}'$ be an ML model for time series forecasting, where $M \in \mathbb{N}$ is the number of static features. A *composition map* is denoted $\mathcal{M}_{\mathcal{S}} : \mathbb{R}^M \to \mathcal{C}_{\mathcal{S}}$ defined by $\mathcal{M}_{\mathcal{S}}(\boldsymbol{x}) = \mathcal{C}_{\mathcal{S}}[g(\boldsymbol{x})]$.

To understand a model $g$, it is crucial to understand its composition map with respect to some meaningful set of motifs. We discuss examples of motifs and when they can be helpful in Appendix A. We define a particular set of motifs that we call *dynamical motifs* (see Table 2). They encode information about the trajectory's first and second derivatives. Moreover, the transition points between these motifs correspond to local minima, maxima, and inflection points. These are the exact properties used in a standard mathematical exercise of *function sketching* whose goal is precisely to understand the function. These motifs form a backbone of TIMEVIEW introduced in Section 5. Dynamical motifs are depicted in Table 2 and defined formally in Example 5 in Appendix A.

Table 2: We introduce a set of *dynamical motifs* that are often important to understand trajectories.

| Symbol | Name | Definition | Visualization |
|--------|------|------------|---------------|
| $s_{+0}$ | Straight line with positive slope | $f(x) = ax + b, a > 0, b \in \mathbb{R}$ | |
| $s_{-0}$ | Straight line with negative slope | $f(x) = ax + b, a < 0, b \in \mathbb{R}$ | |
| $s_{00}$ | Straight line with zero slope | $f(x) = b, b \in \mathbb{R}$ | |
| $s_{++}$ | Increasing and strictly convex | $f'(x) > 0, f''(x) > 0$ | |
| $s_{+-}$ | Increasing and strictly concave | $f'(x) > 0, f''(x) < 0$ | |
| $s_{-+}$ | Decreasing and strictly convex | $f'(x) < 0, f''(x) > 0$ | |
| $s_{--}$ | Decreasing and strictly concave | $f'(x) < 0, f''(x) < 0$ | |

## 5 TIMEVIEW

Based on our formalism in Section 4, we introduce Time series Interpretable Model with Effective VIsualization (TIMEVIEW). This framework consists of two parts: the predictive model based on B-Spline basis functions, and an algorithm for calculating the composition map. This map aims to facilitate model visualization that complements our framework and is demonstrated in Section 7.

**Realizing bi-level transparency through dynamical motifs.** To realize bi-level transparency through dynamical motifs, we need to (1) understand the relation between the feature vectors $\boldsymbol{x}$ and the compositions of the predicted trajectories, and (2) understand the relation between the feature vectors $\boldsymbol{x}$ and the transition points of a given composition. To fulfill these conditions, we need to find a space of trajectories $\hat{\mathcal{Y}}$ satisfying the following criteria.

1. The set of dynamical motifs $\mathcal{S}$ is compatible with the class of predicted trajectories $\hat{\mathcal{Y}}$
2. For every $\hat{y} \in \hat{\mathcal{Y}}$ we can calculate its composition $\mathcal{C}_{\mathcal{S}}[\hat{y}]$

**Cubic splines** are a class of functions that satisfies both criteria mentioned above. We demonstrate that dynamical motifs are compatible with cubic splines in Appendix B. Moreover, it is easy to calculate the dynamical composition of a cubic spline as it is a piece-wise function consisting of cubic polynomials connected at *knots*. We describe the exact procedure below and in Appendix C.

**B-Spline basis functions.** We describe cubic splines as linear combinations of B-Spline (De Boor, 1978) basis functions. Let $\phi_b : [0, T] \to \mathbb{R}$ be a $b^{\text{th}}$ B-Spline basis function of degree 3. Given a set of $B$ basis functions $\{\phi_b\}_{b \in [B]}$, we can express a cubic spline as a linear combination $\hat{y}(t) = \sum_{b=1}^{B} c_b \phi_b(t)$, where $c_b \in \mathbb{R} \ \forall b \in [B]$. Thus, each spline is described by a latent vector $\boldsymbol{c} \in \mathbb{R}^B$.

**Architecture.** To match a feature vector $\boldsymbol{x} \in \mathbb{R}^M$ to a vector $\boldsymbol{c} \in \mathbb{R}^B$ describing a time series, we use an encoder $\boldsymbol{h} : \mathbb{R}^M \to \mathbb{R}^B$. Ultimately, we define our model $g : \mathbb{R}^M \to \hat{\mathcal{Y}}$ as

$$g(\boldsymbol{x})(t) = \hat{y}_{\boldsymbol{x}}(t) = \sum_{b=1}^{B} h(\boldsymbol{x})_b \phi_b(t) \tag{3}$$

**Implementation.** We implement the encoder $\boldsymbol{h}$ as a fully-connected neural network. We choose a set of knots for the B-Spline basis functions based on the training dataset using a heuristic algorithm described in Appendix C (note, the number of knots controls explainability-performance trade-off). The values of $\phi_b$ at times $\boldsymbol{t}^d$ ($\forall b \in [B] \; \forall d \in [D]$) can be efficiently precomputed before the training using `scipy` library's `BSpline` class. We want to minimize the MSE loss between the predicted values of the trajectory $\hat{y}$ at points $\boldsymbol{t}^d$ and the ground truth $\boldsymbol{y}^d$. We also add L2 regularization loss $\mathcal{L}_{\text{L2}}$, so that the B-Spline coefficients (and thus the compositions) do not change too abruptly. The final objective is:

$$\mathcal{L} = \frac{1}{D} \sum_{d=1}^{D} \left( \frac{1}{N_d} \sum_{j=1}^{N_d} \left( y_j^d - \sum_{b=1}^{B} h(\boldsymbol{x}^d)_b \phi_b(t_j^d) \right)^2 \right) + \alpha \mathcal{L}_{\text{L2}}(\boldsymbol{g}) \tag{4}$$

We minimize it using gradient descent. The block diagram describing the training procedure can be seen in Figure 3. Implementation details, including the pseudocode, can be found in Appendix C.

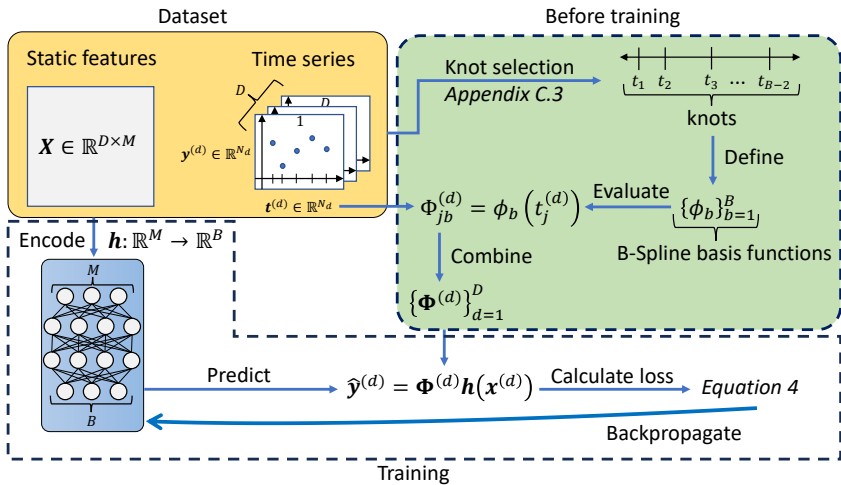

Figure 3: This figure shows a block diagram depicting the training procedure of TIMEVIEW

As with many transparent models (e.g., GAMs, Decision Trees), model visualization is crucial for its interpretability. After TIMEVIEW is trained, we compute the composition map (see Definition 3) and demonstrate how we can visualize it (or a part of it) in Section 7. To compute the composition map, we need to perform *composition extraction* from a predicted trajectory, i.e., calculate $\mathcal{C}_S[\hat{y}]$.

**Composition extraction.** As described earlier, each trajectory is described by a latent vector $\boldsymbol{c} \in \mathbb{R}^B$ and defined as a linear combination of B-Splines, $\hat{y}(t) = \sum_{b=1}^{B} c_b \phi_b(t)$. Each $\phi_b$ is a piece-wise polynomial defined over the intervals determined by the internal knots $(t_1, \ldots, t_{B-2})^4$ chosen by our heuristic algorithm (Appendix C). We can associate a cubic in a monomial basis $(t^3, t^2, t, 1)$ with each of these intervals for each basis function (this can be precomputed). We call these cubics $\psi_{b,k}$, where $k$ ranges from 1 to $B-3$ (the number of intervals). Given a vector $\boldsymbol{c}$, we can now calculate the cubic in a monomial basis for each interval. The $k^{\text{th}}$ interval is just $\sum_{b=1}^{B} c_b \psi_{b,k}$. As it is just a cubic polynomial, we can readily calculate its first and second derivatives and thus assign a composition to the $k^{\text{th}}$ interval. We repeat this process for every other interval, connect all the compositions, and merge some neighboring motifs if they are the same. Ultimately, we get a global composition for the whole $\hat{y}$. See Appendix C for the pseudocode and the block diagram description.

## 6    RELATED WORKS

We explain how our work intersects with related areas of ML. Refer to Appendix F for more details.

**Transparent models for static predictions.** Standard transparent methods for static predictions include linear/logistic regression, scoring systems (Ustun & Rudin, 2016), decision trees/rule lists

---

[4]For B-Splines of degree 3, $B-2$ knots produce $B$ basis functions.

(Angelino et al., 2018; Hu et al., 2019), and generalized additive models (GAMs) (Hastie & Tibshirani, 1986; Lou et al., 2012). Such methods can often be used for time series forecasting by passing the time $t$ as an additional feature. They often satisfy bi-level transparency but have poor performance. In particular, all trajectories predicted by linear regression and GAMs are parallel; thus, they cannot model different trends (Section 7). Decision Trees capture non-additive interactions, enabling flexible forecasting models. However, they require many splits to approximate the ground truth, leading to poor performance or incomprehensibility (Section 7).

**Closed-form expressions.** Symbolic Regression (Schmidt & Lipson, 2009; La Cava et al., 2021) aims to fit closed-form expressions to data, i.e., mathematical formulas composed of a finite number of variables, binary operators $(+, -, \times, \div)$, well-known functions (e.g., $\sin, \exp, \log$), and constants. For instance, $\sin(x^2) - e^{2.1y}$. Differential equations represent another category of mathematical expressions that draw significant interest in the scientific community. Numerous algorithms have been proposed for discovering Ordinary Differential Equations (ODEs) (Brunton et al., 2016; Qian et al., 2022) and Partial Differential Equations (Rudy et al., 2017; Long et al., 2019). Mathematical expressions may not always satisfy bi-level transparency. In fact, the reparametrization of equations to reflect key theoretical quantities is an active area of research (Preacher & Hancock, 2015).

**Feature importance for time series.** While our research focuses on transparent models, many saliency (or feature importance) methods have been developed to highlight which features the model is sensitive to (Ribeiro et al., 2016; Lundberg & Lee, 2017). Although these methods have been extended to time series *inputs* (Crabbé & Schaar, 2021; Leung et al., 2023), limited work has been done to extend them specifically to time series *outputs*. Current XAI techniques either assume the output is a scalar (Siddiqui et al., 2019) (e.g., time series classification (Hao & Cao, 2020)), treat the trajectory as a single object (Gao et al., 2023)—thus do not show *how* a feature changes the trajectory—or show a saliency map at each predicted point separately (Pan et al., 2020), thus allowing only for a bottom-up understanding of the predicted trajectory. The last category also includes many recently proposed methods with attention mechanisms (Alaa & van der Schaar, 2019; Lim et al., 2021). We contrast our framework with feature importance techniques in Appendix E.

**Shapelets and motifs.** As our method discusses the shape of the trajectory, it may seem related to shapelet-based methods (Ye & Keogh, 2009). However, these methods are usually used for data mining and classification tasks. They aim to find subsequences of a time series that represent the most important patterns of each class and thus can be used to distinguish between them (Chen et al., 2022). Similarly, motif discovery identifies short repeating patterns in the time series (Torkamani & Lohweg, 2017) usually for insights into the problem or classification tasks.

## 7 TIMEVIEW IN ACTION

**Answering questions.** Our interactive visualization tool for TIMEVIEW allows for answering questions such as "*What If*", "*How to be that*", and "*How to still be this*" from the XAI Question Bank (Liao et al., 2020) discussed in Section 1. As explained earlier, answering such questions with the current bottom-up approaches may often be futile since the notion of a "different prediction" may be non-interpretable or simplistic. In contrast, TIMEVIEW allows the analysis of a trajectory change at two levels, i.e., the composition of the trajectory or the coordinate of the transition point.

**Visualizing perturbations.** We can visualize the effect of perturbing one or two features at a time using colorful bands (as in Figure 1) and colorful 2D contour plots (Figure 4). In the left panel of Figure 1, we have a movable slider for each feature that changes the predicted trajectory in the center. The colors on the band below the slider signify the composition of the trajectory if the feature is in the corresponding range. This allows us to understand how the *trend* of the trajectory changes as we change each feature (level 1 of bi-level transparency). To understand the properties of this trend (level 2), we choose any of the transition points in the central plot, and we can analyze its position with respect to the chosen feature on the plot on the right. It currently shows how the $y$-coordinate of the second transition point (local minimum) increases as initial tumor volume increases. (Figure 4)

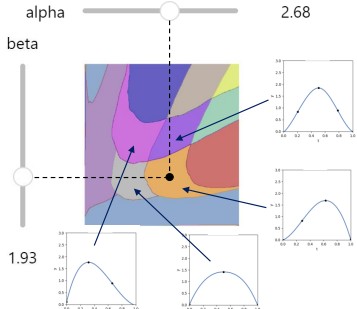

Figure 4: Our interface can visualize perturbing two features simultaneously.

shows how we can visualize the effect of changing two features at a time. Each color in the contour plot corresponds to a different composition, so it is clear how changing the features influences the composition of the trajectory. Please see Appendix E for a more in-depth discussion.

**Comparison with other methods.** In the absence of time series methods fulfilling bi-level transparency, we adapt static transparent methods, such as linear regression, decision trees, and GAMs (Lou et al., 2012; Nori et al., 2019) to time series forecasting by treating time as a feature, denoted as Linear-T, DecisionTree-T, and GAM-T. We also compare with methods discovering closed-form expressions for trajectories, such as PySR for symbolic regression (Cranmer, 2020), and SINDy for ODE discovery (Brunton et al., 2016). We also include black-box models RNN, $\Delta t$-RNN, and state-of-the-art tree-based models adapted to time series forecasting (XGB-T (Chen & Guestrin, 2016), LGBM-T (Ke et al., 2017), CatBoost-T (Prokhorenkova et al., 2018)).

Experiments were conducted on four real-world datasets (Airfoil (Brooks et al., 1989), flchain (Dispenzieri et al., 2012), Stress-Strain (Aakash et al., 2019), and Tacrolimus (Woillard et al., 2011)) and three synthetic ones (Sine, Beta, and Tumor, the latter based on a model from (Wilkerson et al., 2017)). The synthetic datasets are constructed to contain trajectories exhibiting many different trends. Figure 1, Figure 4, Figure 1 show TIMEVIEW fitted to Sine, Beta, and Tumor datasets. As shown in Table 3, TIMEVIEW outperforms the transparent methods and closed-form expression on most datasets and achieves comparable performance to the black boxes. Details about the experiments can be found in Appendix D.

Table 3: Comparison between TIMEVIEW, other transparent methods, closed-form expressions, and black boxes. The numbers denote a mean squared error. The lower, the better. **Boldface red** denotes the best black box results, **boldface orange** denotes the best closed-form expression results, and **boldface green** denotes the best transparent results. † denotes failure to converge. Note, RNN only works for regular time series.

| Method | Real datasets | | | | Synthetic datasets | | |
|---|---|---|---|---|---|---|---|
| | Airfoil | flchain | Stress-Strain | Tacrolimus | Tumor | Sine | Beta |
| **Black boxes** | | | | | | | |
| RNN | - | $0.26 \pm 0.02$ | - | - | $0.02 \pm 0.01$ | $0.02 \pm 0.01$ | $0.02 \pm 0.02$ |
| $\Delta t$-RNN | $0.17 \pm 0.02$ | $0.27 \pm 0.01$ | $0.14 \pm 0.01$ | $0.41 \pm 0.05$ | $0.02 \pm 0.01$ | $0.02 \pm 0.01$ | $0.02 \pm 0.02$ |
| XGB-T | $\mathbf{0.09 \pm 0.00}$ | $\mathbf{0.20 \pm 0.00}$ | $\mathbf{0.02 \pm 0.00}$ | $\mathbf{0.29 \pm 0.01}$ | $0.01 \pm 0.00$ | $0.53 \pm 0.00$ | $0.07 \pm 0.00$ |
| LGBM-T | $0.11 \pm 0.00$ | $\mathbf{0.20 \pm 0.00}$ | $0.08 \pm 0.00$ | $\mathbf{0.29 \pm 0.00}$ | $0.01 \pm 0.00$ | $\mathbf{0.00 \pm 0.00}$ | $\mathbf{0.00 \pm 0.00}$ |
| CatBoost-T | $\mathbf{0.09 \pm 0.01}$ | $0.21 \pm 0.00$ | $0.05 \pm 0.00$ | $0.37 \pm 0.04$ | $\mathbf{0.00 \pm 0.00}$ | $\mathbf{0.00 \pm 0.00}$ | $\mathbf{0.00 \pm 0.00}$ |
| **Closed-form expressions** | | | | | | | |
| PySR | $\mathbf{0.26 \pm 0.03}$ | $\mathbf{0.22 \pm 0.01}$ | $\mathbf{0.48 \pm 0.11}$ | $\mathbf{0.36 \pm 0.04}$ | $0.10 \pm 0.02$ | $\mathbf{0.05 \pm 0.04}$ | $\mathbf{0.24 \pm 0.03}$ |
| SINDy | $0.61 \pm 0.00$ | $0.39 \pm 0.00$ | † | $1.11 \pm 0.00$ | $\mathbf{0.07 \pm 0.00}$ | $1.30 \pm 0.00$ | $2.74 \pm 0.00$ |
| **Transparent models** | | | | | | | |
| Linear-T | $0.37 \pm 0.00$ | $0.34 \pm 0.00$ | $0.66 \pm 0.00$ | $0.57 \pm 0.00$ | $0.68 \pm 0.00$ | $0.99 \pm 0.00$ | $1.03 \pm 0.00$ |
| DecisionTree-T | $0.36 \pm 0.00$ | $\mathbf{0.21 \pm 0.00}$ | $0.15 \pm 0.00$ | $\mathbf{0.31 \pm 0.00}$ | $0.22 \pm 0.00$ | $0.10 \pm 0.00$ | $0.34 \pm 0.00$ |
| GAM-T | $0.28 \pm 0.01$ | $0.32 \pm 0.00$ | $0.09 \pm 0.00$ | $0.38 \pm 0.00$ | $0.54 \pm 0.00$ | $0.54 \pm 0.00$ | $0.69 \pm 0.00$ |
| TIMEVIEW | $\mathbf{0.13 \pm 0.01}$ | $0.24 \pm 0.02$ | $\mathbf{0.04 \pm 0.00}$ | $\mathbf{0.31 \pm 0.03}$ | $\mathbf{0.00 \pm 0.00}$ | $\mathbf{0.02 \pm 0.00}$ | $\mathbf{0.04 \pm 0.00}$ |

## 8 DISCUSSION AND CONCLUSION

**Applications.** We believe bi-level transparency and our mathematical framework can inspire future XAI methods. For instance, note, that models adhering to our framework (like TIMEVIEW) provide additional output next to the standard forecasted trajectory: the current composition and the coordinates of the transition points. Traditional XAI techniques for regression and classification can be applied to these additional outputs, instead of individual trajectory points, to gain more meaningful explanations. Thus, techniques such as feature importance methods (Lundberg & Lee, 2017), local surrogates (Ribeiro et al., 2016), and counterfactual explanations (Karimi et al., 2020) can now be extended to time series forecasting settings. These, in turn, can open domains where the applicability of ML has been limited due to transparency concerns, including medicine, finance, and science.

**Limitations and open challenges.** TIMEVIEW is a particular application of bi-level transparency for time series forecasting from static features. We hope future works will extend it to settings where the input may contain the previous part of the trajectory or other exogenous time series (further discussion in Appendix E).

**Ethics statement.** In this paper, we present a novel conceptual framework for enhancing transparency in the domain of time series forecasting, accompanied by its practical implementation known as TIMEVIEW. A better understanding of machine learning models serves critical purposes such as model debugging and identifying and mitigating potential harmful biases. However, XAI techniques can also be misused to foster unwarranted trust in models or to merely achieve surface-level compliance with regulatory standards. As highlighted in our paper, domains such as medicine and pharmacology involve high-stakes scenarios. Therefore, prior to deploying our model in such contexts, a rigorous examination is imperative to ensure it does not endorse decisions that could prove detrimental to individuals' well-being.

**Reproducibility statement.** All mathematical definitions are provided in Section 4 and Appendix A. The proofs of theoretical results are shown in Appendix B. The implementation, including block diagrams and pseudocode, is discussed in Section 5 and in Appendix C. The experiment settings are discussed in Section 7 and Appendix D. The code to reproduce the results and for the visualization tool can be found at `https://github.com/krzysztof-kacprzyk/TIMEVIEW` and at the wider lab repository `https://github.com/vanderschaarlab/TIMEVIEW`.

**Acknowledgments.** This work was supported by Roche and AstraZeneca. We want to thank Katarzyna Kobalczyk, Fergus Imrie, Andrew Rashbass, and anonymous reviewers for their useful comments and feedback on earlier versions of this work.

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

TABLE OF SUPPLEMENTARY MATERIALS

## A  NOTATION AND DEFINITIONS

### A.1  NOTATION

We present the symbols used in this work and their meanings in Table 4.

Table 4: Notation used throughout the paper

| Symbol | Meaning |
| --- | --- |
| $[n]$ | $\{1, 2, \ldots, n\}$ |
| $I$ | Interval (Definition 4) |
| $f\|_I$ | Restriction of $f$ onto $I$, i.e., $f\|_{I_0} : I_0 \to \mathbb{R}$, $f\|_{I_0}(t) = f(t) \ \forall t \in I_0$. |
| $\mathcal{F}$ | Interval functions (Definition 6) |
| $I + c$ | Shifted interval (Definition 4) |
| $f\|I \sim s$ | $f$ on $I$ has motif $s$ (Definition 1) |
| $\mathcal{S}$ | Set of motifs (Definition 1) |
| $\mathcal{C}_{\mathcal{S}}$ | Set of compositions for a given set of motifs $\mathcal{S}$ (Definition 2) |
| $\mathcal{C}_{\mathcal{S}}[f]$ | Composition of $f$ in $\mathcal{S}$ (Definition 2) |
| $\mathcal{M}_{\mathcal{S}}$ | Composition map for a set of motifs $\mathcal{S}$ (Definition 3) |
| $\phi$ | B-Spline basis function (Section 5) |
| $\boldsymbol{h}$ | Encoder (Section 5) |
| $g$ | TIMEVIEW model (Section 5) |
| $\hat{y}$ | Predicted trajectory (Section 3) |
| $\hat{\mathcal{Y}}$ | Set of predicted trajectories (Section 3) |
| $D$ | Number of samples |
| $M$ | Number of static features |
| $N_d$ | Number of measurements for a sample $d$ |
| $B$ | Number of basis functions |

### A.2  GLOSSARY OF TERMS

In Table 5, we present a glossary of important words and where they can be found in the paper.

### A.3  DEFINITIONS

**Definition 4** (Interval). We say $I$ is an interval (of $\mathbb{R}$) if it is an open interval, closed interval, or half-closed interval. The interval has to contain more than one point. We denote the set of all intervals on $\mathbb{R}$ as $\mathcal{I}$. Let $c \in \mathbb{R}$, we denote the shifted interval as $I + c = \{x + c \mid x \in I\}$.

**Definition 5** (Interval partition). Interval partition of an interval $I$ is a sequence of intervals $(I_1, \ldots, I_n)$ such that $\bigcup_{i=1}^{n} I_i = I$, $I_i \cap I_{i+1} = \emptyset \ \forall i \in [n-1]$ and the right boundary of $I_i$ should be no greater than the left boundary of $I_{i+1}$ for all $i \in [n-1]$.

**Definition 6** (Interval function). Let $I \subset \mathbb{R}$ be any interval, we call any function $f : I \to \mathbb{R}$ an *interval function* and we denote its domain as $\mathrm{dom}(f)$. We denote the set of all interval functions as $\mathcal{F}$.

Table 5: All important words defined in the paper and their locations

| Word | Location in the paper |
|---|---|
| Bi-level transparency | Section 2 |
| Compatibility | Definition 9 |
| Composition | Definition 2 |
| Composition extraction | Section 5 |
| Composition map | Definition 3 |
| Dynamical motifs | Example 5 |
| Exclusive motifs | Definition 12 |
| Inclusive motifs | Definition 11 |
| Interval | Definition 4 |
| Interval function | Definition 6 |
| Interval partition | Definition 5 |
| Motif | Definition 1 |
| Property (of a trend) | Section 2 |
| TIMEVIEW | Section 5 |
| Transition point | Definition 2 |
| Trend | Section 2 |

**Definition 7** (Motif). A motif $s$ is a binary relation between the set of interval functions $\mathcal{F}$ and the set of intervals $\mathcal{I}$ (i.e., $s \in \mathcal{F} \times \mathcal{I} = \{(f, I) \mid f \in \mathcal{F},\ I \in \mathcal{I}\}$). We denote $(f, I) \in s$ as $f|I \sim s$ and read it as "$f$ on $I$ has a motif $s$". Each motif $s$ needs to be:

- *well-defined*, i.e., for any $f \in \mathcal{F}$, and any $I \in \mathcal{I}$,

$$f|I \sim s \implies I \subseteq \mathrm{dom}(f) \tag{5}$$

- *translation-invariant*, i.e., for any $I \in \mathcal{I}$, and any $f \in \mathcal{F}$,

$$f|I \sim s \iff f \circ (x - c)|(I + c) \sim s\ \forall c \in \mathbb{R} \tag{6}$$

**Definition 8** (Composition). Let $f : I \to \mathbb{R}$ be an interval function and $\mathcal{S}$ be a set of motifs. A *motif sequence* of $f$ in $\mathcal{S}$ is a finite sequence of motifs $(s_1, \ldots, s_d)$, such that there exists an interval partition $(I_1, \ldots, I_d)$ of $I$ such that $f|I_j \sim s_j\ \forall j \in [d]$. A *composition* of $f$ in $\mathcal{S}$ is the shortest *motif sequence* of $f$ in $\mathcal{S}$. The points between the intervals are called the *transition points*. The set of all compositions for a given set of motifs $\mathcal{S}$ is denoted by $\mathcal{C}_\mathcal{S}$. A set of motifs $\mathcal{S}$ is called *compatible* with a subset $\mathcal{F}' \subset \mathcal{F}$ if for every $f \in \mathcal{F}'$ there exists a unique composition, denoted $\mathcal{C}_\mathcal{S}[f]$

**Definition 9** (Compatibility). Let $\mathcal{S}$ be a set of motifs and $\mathcal{F}' \subset \mathcal{F}$ be a subset of interval functions. $\mathcal{S}$ is called *compatible* with $\mathcal{F}'$ if for every $f \in \mathcal{F}'$ there exists a unique composition, $\mathcal{C}_\mathcal{S}[f]$ and a unique sequence of the transition points associated with this composition.

**Definition 10** (Composition map). Let a set of motifs $\mathcal{S}$ be compatible with some subset $\mathcal{F}' \subset \mathcal{F}$. Let $g : \mathbb{R}^M \to \mathcal{F}'$ be a machine learning model for time series forecasting. A *composition map* is denoted $\mathcal{M}_\mathcal{S} : \mathbb{R}^M \to \mathcal{C}_\mathcal{S}$ defined by $\mathcal{M}_\mathcal{S}(\boldsymbol{x}) = \mathcal{C}_\mathcal{S}[g(\boldsymbol{x})]$.

**Definition 11** (Inclusive motifs). We say that a motif $s$ is *inclusive* if for any function $f \in \mathcal{F}$ and any interval $I \in \mathcal{I}$, we have

$$f|I \sim s \implies f|I_0 \sim s\ \forall I_0 \subset I \tag{7}$$

**Definition 12** (Exclusive motifs). We say that a finite set of motifs $\mathcal{S}$ is *exclusive* if for any function $f \in \mathcal{F}$ and any interval $I \in \mathcal{I}$, we have

$$f|I \sim s \implies f|I \not\sim s'\ \forall s \neq s' \tag{8}$$

for all $s, s' \in \mathcal{S}$. In other words, if $f$ is in a particular motif on the interval $I$ then it cannot be in any other motif on this interval.

### A.4 MOTIF EXAMPLES

**Example 1** (Range motifs). Let $a, b \in \mathbb{R}$ such that $a < b$. We define *range* motifs $s_<, s_=, s_>$ as:

- $f|I \sim s_<$ if $\forall x \in I\ f(x) < a$
- $f|I \sim s_=$ if $\forall x \in I\ a \le f(x) \le b$
- $f|I \sim s_>$ if $\forall x \in I\ b < f(x)$

These motifs are beneficial when our interest lies not in the trajectory's exact shape, but in whether its value falls within a specific range $[a, b]$ and at what point in time. This can be particularly useful in control problems where different policies apply based on whether a variable is within or outside the operating range $[a, b]$. More motif examples can be found in Appendix A.

**Example 2** (Threshold motifs). Range motifs (Example 1) can be modified by choosing $a = b$. Then we end up with only two motifs:

- $f|I \sim s_<$ if $\forall x \in I\ f(x) < a$
- $f|I \sim s_>$ if $\forall x \in I\ a < f(x)$

These motifs are useful for monitoring whether the trajectory is above or below a certain threshold.

**Example 3** (Monotonic motifs). Let us define *monotonic* motifs $s_+ s_-, s_0$ which are defined as:

- $f|I \sim s_+$ if $f$ is strictly increasing on $I$ ($\forall x, y \in I\ x < y \implies f(x) < f(y)$)
- $f|I \sim s_-$ if $f$ is strictly decreasing on $I$ ($\forall x, y \in I\ x < y \implies f(x) > f(y)$)
- $f|I \sim s_0$ if $f$ is constant on $I$ ($\forall x, y \in I\ f(x) = f(y)$)

**Example 4** (Derivative range motifs). Let $a, b \in \mathbb{R}$ and $a < b$. We define *derivative range motifs* as:

- $f|I \sim s_<$ if $f|_I \in \mathcal{C}^1$ and $\forall x \in I\ f'(x) < a$
- $f|I \sim s_=$ if $f|_I \in \mathcal{C}^1$ and $\forall x \in I\ a \le f'(x) \le b$
- $f|I \sim s_>$ if $f|_I \in \mathcal{C}^1$ and $\forall x \in I\ b \le f'(x)$

**Example 5** (Dynamical motifs). We define seven dynamical motifs: $s_{0+}, s_{0-}, s_{00}, s_{++}, s_{+-}, s_{-+}, s_{--}$ as follows (also visualized in Table 2).

- $f|I \sim s_{+0}$ if $\exists a > 0\ \exists b \in \mathbb{R}\ \forall x \in I\ f(x) = ax + b$
- $f|I \sim s_{-0}$ if $\exists a < 0\ \exists b \in \mathbb{R}\ \forall x \in I\ f(x) = ax + b$
- $f|I \sim s_{00}$ if $\exists b \in \mathbb{R}\ \forall x \in I\ f(x) = b$
- $f|I \sim s_{++}$ if $f|_I \in \mathcal{C}^2$ and $\forall x \in \text{int}(I)\ f'(x) > 0, f''(x) > 0$
- $f|I \sim s_{+-}$ if $f|_I \in \mathcal{C}^2$ and $\forall x \in \text{int}(I)\ f'(x) > 0, f''(x) < 0$
- $f|I \sim s_{-+}$ if $f|_I \in \mathcal{C}^2$ and $\forall x \in \text{int}(I)\ f'(x) < 0, f''(x) > 0$
- $f|I \sim s_{--}$ if $f|_I \in \mathcal{C}^2$ and $\forall x \in \text{int}(I)\ f'(x) < 0, f''(x) < 0$

# B THEORETICAL RESULTS

**Theorem 1** (Dynamical motifs are compatible with cubic splines). *Let $\mathcal{S}$ be the set of dynamical motifs and let $\mathcal{F}'$ be the set of cubic splines on some interval $[a, b]$. Then $\mathcal{S}$ is compatible with $\mathcal{F}'$*

Before we prove this theorem, we will prove two propositions about dynamical motifs, and two helpful lemmas.

**Proposition 1.** *Dynamical motifs are inclusive (as defined in Definition 11.*

*Proof.* Observe that $s_{+0}$, $s_{-0}$, and $s_{00}$ can be equivalently defined as:

- $f|I \sim s_{+0}$ if $f|_I \in \mathcal{C}^2$ and $\forall x \in \text{int}(I) f'(x) > 0, f''(x) = 0$
- $f|I \sim s_{-0}$ if $f|_I \in \mathcal{C}^2$ and $\forall x \in \text{int}(I) f'(x) < 0, f''(x) = 0$
- $f|I \sim s_{00}$ if $f|_I \in \mathcal{C}^2$ and $\forall x \in \text{int}(I) f'(x) = 0, f''(x) = 0$

This (with the rest of the motifs defined in Example 5) shows that each motif is uniquely defined by the signs of its first and second derivatives.

If $\text{sign}(f')$ on an interval $I$ is constant then $\text{sign}(f'|_{I_0}) = \text{sign}(f')$ for any interval $I_0 \subset I$. The same holds for the second derivative. This proves that dynamical motifs are inclusive. $\square$

**Proposition 2.** *Dynamical motifs are exclusive (as defined in Definition 12.*

*Proof.* As we shown in the proof of Proposition 1, each dynamical motif is uniquely defined by the signs of its first and second derivatives. If $\text{sign}(f')$ on an interval $I$ is constant then it cannot be equal to anything else. The same holds for the second derivative. This proves that dynamical motifs are exclusive. □

**Lemma 1** (Uniqueness of dynamical motif sequences). *Let $\mathcal{S}$ be the set of dynamical motifs and let $f$ be an interval function. If there exists a motif sequence for $f$, $(s_1, \ldots, s_d)$, satisfying $s_i \in \mathcal{S} \forall i \in [d]$, $s_i \neq s_{i+1} \forall i \in [d-1]$, then this motif sequence is unique.*

*Proof.* Consider two motif sequences $(s_1, \ldots, s_d)$, $(p_1, \ldots, p_e)$ such that for both of them every two consecutive motifs are different. We show that these sequences need to be the same.

Let us compare $p_1$ and $s_1$. Consider the intervals described by $s_1$ and $p_1$ and denote them $I_1$ and $J_1$ respectively. By inclusivity of dynamical motifs (Proposition 1), $f|(I_1 \cap J_1) \sim s_1$ and $f|(I_1 \cap J_1) \sim p_1$. That implies, by exclusivity (Proposition 2), that $s_1 = p_1$. That also shows that $J_1$ cannot be bigger than $I_1$. If that were the case, it would overlap with $I_2$ (described by $s_2$). But $s_2 \neq s_1 = p_1$. So, $J_1 \subset I_1$. If $J_1$ is smaller than $I_1$ then $J_2 \subset (I_1 \setminus J_1)$, where $J_2$ is described by $p_2$, and $p_2 = s_1 = p_1$. But we assumed that $p_1 \neq p_2$. Thus, $J_1 = I_1$. Now, we can apply exactly the same reasoning to the next pair of motifs $(s_2, p_2)$. We continue this procedure and in the end, we get these two motif sequences are the same. Note, it is also possible that $I_1 \setminus J_1$ is just a single point. That is why the intervals corresponding to the motif sequence are unique up to the boundaries of the interval, i.e., the intervals might differ by single points. □

**Lemma 2** (Dynamic motifs are compatible with cubics). *Let $\mathcal{S}$ be the set of dynamical motifs and let $\mathcal{Q}$ be the set of polynomials up to a third degree on $\mathbb{R}$. Then $\mathcal{S}$ is compatible with $\mathcal{Q}$.*

*Proof.* Consider any polynomial $f \in \mathcal{Q}$. Let's write $f$ as $f(t) = at^3 + bt^2 + ct + d$. We can calculate its first and second derivatives. Respectively, $f'(t) = 3at^2 + 2bt + c$, $f''(t) = 6at + 2b$. $f'$ and $f''$ divide $\mathbb{R}$ into intervals where on each interval $f'$ (or $f''$ respectively) is either positive, negative, or equal to zero. $f'$ divides $\mathbb{R}$ into at most 3 intervals, and $f''$ divides $\mathbb{R}$ into at most two intervals. There might be some isolated points that do not belong to any of these intervals. For instance, if $f''(t) = t$, we have two intervals: $(-\infty, 0)$ where $f''$ is negative, $(0, +\infty)$ where $f''$ is positive, and an isolated point at $t = 0$. For $f''$ we have at most one isolated point, and for $f'$ we have at most two isolated points. We can now take the intersection of every interval defined by $f'$ with every other interval defined by $f''$. This gives us at most 4 intervals where each interval belongs to one of 9 different configurations that we denote by two symbols. The first symbol denotes the sign of the first derivative $(+, -, 0)$, and the second sign denotes the sign of the second derivative. For instance, an interval might be described as $-+$ if it was created by intersecting an interval that has a negative first derivative with an interval that has a positive second derivative. We observe that two configurations are impossible, $0+$ and $0-$, so we end up with 7 possible configurations.

We observe that if an interval $I$ is described by $+-$ then by Example 5 $f|I \sim s_{+-}$. Similarly for $++$, $-+$, and $--$. If an interval $I$ is denoted by $00$ then it means that $6at + 2b = 0 \ \forall t \in I$, and $3at^2 + 2bt + c = 0 \ \forall t \in I$. From the first equation, we get that $a = 0, b = 0$, and from the second one we get $c = 0$. That means that $f|I \sim s_{00}$. Similarly, an interval $I$ is described by $+0$ then $a = 0, b = 0$ and $c > 0$. That means $f$ is given by $ct + d$, where $c > 0$, so $f|I \sim s_{+0}$. Analogously with $-0$. The isolated points mentioned above are exactly the transition points between different motifs.

This shows how for every polynomial up to a third degree, there exists a sequence of dynamical motifs and we showed explicitly how to construct such a sequence. Let us call this sequence $(s_1, \ldots, s_d)$. Observe that the shortest motif sequence for $f$ cannot have two identical consecutive motifs. If that were the case, they could have been combined into a shorter motif sequence. As every two consecutive motifs of $(s_1, \ldots, s_d)$ are different then, by Lemma 1, this is the only such sequence and thus it is the shortest. So $(s_1, \ldots, s_d)$ is a composition. That shows that for every $f \in \mathcal{Q}$ there exists a unique composition and thus $\mathcal{S}$ is compatible with $\mathcal{Q}$. □

Now, we can prove Theorem 1.

*Proof of theorem Theorem 1.* Let $f$ be a cubic spline defined on internal knots $t_1, \ldots, t_n$. Let us denote $f_i$ the polynomial defined on $(t_i, t_{i+1})$. To construct a composition for $f$, we construct compositions for each of the polynomials $f_i$ (as described in Lemma 2). Then we look at the neighboring motifs on both sides of every knot. If the motifs are the same, then we combine them and remove the transition point. This procedure gives us a motif sequence for $f$ that we denote as $(s_1, \ldots, s_d)$.

Observe that the shortest motif sequence for $f$ cannot have two identical consecutive motifs. If that were the case, they could have been combined into a shorter motif sequence (this is true because cubic splines have continuous first and second derivatives at the knots). As every two consecutive motifs of $(s_1, \ldots, s_d)$ are different then, by Lemma 1, this is the only such sequence and thus it is the shortest. So $(s_1, \ldots, s_d)$ is a composition. That shows that for every cubic spline, there exists a unique composition, and thus $\mathcal{S}$ is compatible with cubic splines. $\square$

## C  IMPLEMENTATION

### C.1  MODEL

**Pseudocode.** The pseudocode of the model training in TIMEVIEW is shown in Algorithm 1.

---

**Algorithm 1** TIMEVIEW Model training

---

**Input:** Static features $\boldsymbol{X} \in \mathbb{R}^{D \times M}$
**Input:** Time series $\{\boldsymbol{t}^{(d)}\}_{d=1}^{D}, \{\boldsymbol{y}^{(d)}\}_{d=1}^{D}, \boldsymbol{t}^{(d)} \in \mathbb{R}^{N_d}, \boldsymbol{y}^{(d)} \in \mathbb{R}^{N_d}$
**Input:** Number of basis functions $B$
**Input:** Gradient-based optimization algorithm $\mathcal{O}$
**Output:** Trained model $G : \mathbb{R}^M \to \hat{\mathcal{Y}}$

$\quad t_1, \ldots, t_{B-2} \leftarrow \text{SELECTKNOTS}(\{\boldsymbol{t}^{(d)}\}_{d=1}^{D}, \{\boldsymbol{y}^{(d)}\}_{d=1}^{D})$ $\qquad\qquad\qquad$ ▷ Appendix C.3
$\quad \{\phi_b\}_{b=1}^{B} \leftarrow$ B-Spline basis functions for knots $t_1, \ldots, t_{B-2}$
$\quad$ Initialize matrices $\{\boldsymbol{\Phi}^{(d)}\}_{d=1}^{D}, \boldsymbol{\Phi}^{(d)} \in \mathbb{R}^{N_d \times B}$
$\quad \Phi_{jb}^{(d)} \leftarrow \phi_b(t_j^{(d)})$
$\quad$ **procedure** LOSS($\boldsymbol{h}$)
$\quad\quad \hat{\boldsymbol{y}}^{(d)} \leftarrow \boldsymbol{\Phi}^{(d)} \boldsymbol{h}(\boldsymbol{x}^{(d)})$
$\quad\quad \mathcal{L} \leftarrow \frac{1}{D} \sum_{d=1}^{D} \left( \frac{1}{N_d} \sum_{j=1}^{N_d} \left( \hat{y}_j^{(d)} - y_j^{(d)} \right)^2 \right) + \alpha \mathcal{L}_{L2}$
$\quad\quad$ **return** $\mathcal{L}$
$\quad$ **end procedure**
$\quad \boldsymbol{h} = \mathcal{O}(\text{LOSS})$ $\qquad\qquad\qquad\qquad\qquad\qquad\qquad\qquad\qquad\qquad\qquad\qquad$ ▷ Training
$\quad$ **procedure** G($\boldsymbol{x} \in \mathbb{R}^M, \boldsymbol{t} \in \mathbb{R}^N$)
$\quad\quad$ Initialize $\boldsymbol{\Phi} \in \mathbb{R}^{N \times B}$
$\quad\quad \Phi_{jb} \leftarrow \phi_b(t_j)$
$\quad\quad \hat{\boldsymbol{y}} \leftarrow \boldsymbol{\Phi} \boldsymbol{h}(\boldsymbol{x})$
$\quad\quad$ **return** $\hat{\boldsymbol{y}}$
$\quad$ **end procedure**
$\quad$ **return** G

---

**Architecture of the encoder.** We implement encoder $\boldsymbol{h}$ as a fully connected neural network with 3 hidden layers. We also include dropout (Srivastava et al., 2014) and batch normalization (Ioffe & Szegedy, 2015). The sizes of the hidden layers, activation functions, and dropout probability are fine-tuned.

**Dynamic bias.** We add an additional constant basis function (equal to 1 everywhere) as a "bias" that is adjusted for each sample.

**Hyperparameters.** We perform hyperparameter tuning using Optuna (Akiba et al., 2019) and run it for 100 trials. We describe the hyperparameters we tune and their ranges in Table 6. We fix the number of basis functions to be 9 for all real datasets and Tumor, and 5 for Sine and Beta datasets. We found that 9 usually gives the best or nearly the best performance while maintaining compositions that are still short. We chose 5 for Sine and Beta datasets because the compositions

of trajectories in these datasets are very short, so there was no need for a bigger number of basis functions. In practice, this parameter can be fine-tuned but numbers between 5 and 9 work well by default.

Table 6: TIMEVIEW hyperparameter ranges

| Hyperparameter | Considered values |
|---|---|
| Hidden sizes | Integers from $[16, 128]$ |
| Activation | {ReLU,Sigmoid,Tanh,Leaky ReLU,ELU,SeLU} |
| Dropout probability | Float from $[0.0, 0.5]$ |
| Learning rate | Float (log) from $[0.0001, 0.1]$ |
| Batch size | {64,128} |
| Weight decay | Float (log) from $[0.000001, 0.1]$ |

## C.2 COMPOSITION EXTRACTION

**Block diagram.** The composition extraction procedure in TIMEVIEW is depicted in Algorithm 2.

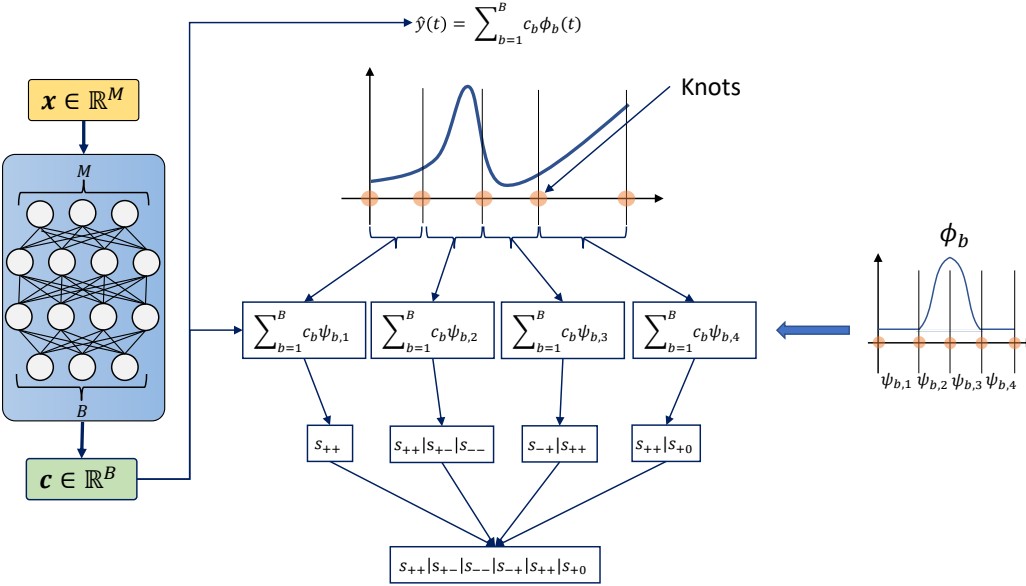

Figure 5: This figure shows a block diagram depicting TIMEVIEW composition extraction

**Pseudocode.** The pseudocode of composition extraction implemented in TIMEVIEW is shown in Algorithm 2.

## C.3 ALGORITHM FOR KNOT SELECTION

Each B-Spline basis function is determined by its knots—places where two polynomials meet. The challenge of knot selection in TIMEVIEW comes from the necessity of having fixed knots for all trajectories. As each trajectory may have a different optimal knot placement, we aim to find knots that may not be optimal but are nevertheless chosen based on the trajectories in the dataset. For instance, there is no need to have many knots in parts of the domain where the trajectories vary little. But it might be beneficial to have more knots in parts of the domain where the trajectories do vary. To achieve this goal we propose the following heuristic algorithm based on the `UnivariateSpline` function in `scipy` library (Virtanen et al., 2020).

---

**Algorithm 2** TIMEVIEW Composition Extraction

---

**Input:** Static features $\boldsymbol{x} \in \mathbb{R}^M$
**Input:** Encoder $\boldsymbol{h}$
**Input:** Knots $t_1, \ldots, t_{B-2}$
**Input:** Method for calculating the composition of a cubic COMPOSITIONCUBIC
**Output:** Composition $\mathcal{C}$
   $\{\phi_b\}_{b=1}^B \leftarrow$ B-Spline basis functions for knots $t_1, \ldots, t_{B-2}$
   $\{\psi_{bk}\}_{b\in[B], k\in[B-3]} \leftarrow \phi_b$ on interval $[t_k, t_{k+1}]$ is given by a cubic $\psi_{bk}$
   $\boldsymbol{c} = \boldsymbol{h}(\boldsymbol{x})$
   $\mathcal{C}_{\text{list}} \leftarrow$ empty list
   **for** $k = 1$ to $B - 3$ **do**
      $\mathcal{C}_{\text{list}} \leftarrow$ append COMPOSITIONCUBIC$\left(\sum_{b=1}^B c_b \phi_{bk}, t_k, t_{k+1}\right)$
   **end for**
   $\mathcal{C} \leftarrow$ concatenate $\mathcal{C}_{\text{list}}$
   **for** $i = 1$ to $\texttt{len}(\mathcal{C}) - 1$ **do**
      **if** $\mathcal{C}_i == \mathcal{C}_{i+1}$ **then**
         remove $\mathcal{C}_i$
      **end if**
   **end for**
   **return** $\mathcal{C}$

---

`UnivariateSpline` function takes the trajectory (both $t$ and $y$), a smoothing parameter $s$[5], calculates the optimal number of knots, and returns their placement. For each trajectory in our training dataset, we test different values of $s$ to find the one that gives the exact number of knots we want (using a method similar to binary search). Then we take the positions of these knots and add them to our set of found knots. We repeat this procedure for every trajectory. Then we use the K-means clustering algorithm from `scikit-learn` library (Pedregosa et al., 2011) with the number of clusters equal to the desired number of knots. The clusters returned by the algorithm are the selected knots.

# D  EXPERIMENTS

## D.1  DATASETS

All datasets are split into training, validation, and testing sets with ratios $(0.7 : 0.15 : 0.15)$.

### D.1.1  REAL DATASETS

**Airfoil.** Airfoil Self-Noise dataset (Brooks et al., 1989) is a UCI dataset, obtained from a series of aerodynamics and acoustic tests of airfoil blade sections conducted in a wind tunnel. The static features are the angle of attack, chord length, free-stream velocity, and suction side displacement thickness. The output is a sound pressure level at specified frequencies (obtained by a Fourier transform of the raw signal). We treat the sound pressure level with respect to the frequency as a time series, i.e., $t$—frequency, $y$—sound pressure level. As the frequencies are uniformly chosen on a logarithmic scale, we take the log transform of $t$. The processed dataset contains 106 samples, each having on average 14 measurements.

**flchain.** Flchain is a dataset of the subjects from a study of the relationship between serum-free light chain and mortality (Dispenzieri et al., 2012), often used in time-to-event problems. We train a Random Survival Forest using `scikit-survival` library (Pölsterl, 2020). Then for each sample from the dataset, we generate a survival function and sample it at 20 predetermined points. The static features are the same as in the original dataset, i.e., age, serum creatinine, the FLC group, kappa portion of serum FLC, lambda portion of FLC, monoclonal gammapothy (MGUS), and sex. The time series is described by $y$—the probability of survival up to this point, $t$—time in days. We subsample the dataset to get 1000 samples, each with 20 measurements.

---

[5]Note, do not confuse with $s$ denoting motifs in our work

**Stress-Strain.** We take the stress-strain curves of aluminum 6061-T651 obtained by Aakash et al. (2019). The aluminum samples are sourced from 9 different lots at a few temperatures between 20°C and 300°C. W treat the temperature and lot as static features and the time series is defined by $t$—strain, $y$—experienced stress. After preprocessing, we end up with 100 samples, each with 212 measurements on average.

**Tacrolimus.** We take the drug concentration curves of two tacrolimus formulations obtained by Woillard et al. (2011). The static features include sex, weight, hematocrit, hemoglobin, creatinine, dose, CYP3A5 genotype, and formulation. We also supplement it with the initial concentration of the drug in the blood. After preprocessing, we end up with 90 samples, each with 10 measurements on average.

### D.1.2 SYNTHETIC DATASETS

**Tumor.** We take the tumor growth model proposed in Wilkerson et al. (2017). This model is described by the following equation:

$$y(t) = \phi \exp(-dt) + (1 - \phi) \exp(gt) - 1 \tag{9}$$

Where $\phi, d, g$ are free parameters corresponding to the proportion of tumor cells that undergo cell death due to treatment, decay of the tumor, and the growth of the tumor. This equation has been shown to capture many different trends of tumor volume over time and that is why we decided to use it in our experiments. To make the model more realistic we multiply $y$ by the initial tumor volume, and we add the following static features: age, weight, and drug dosage. We describe the parameters from Equation (9) using the following relationships:

$$\begin{aligned} g &= g_0 * (age/20.0)^{0.5} \\ d &= d_0 * dosage/weight \\ \phi &= 1/(1 + \exp(-dosage * \phi_0)) \end{aligned} \tag{10}$$

These relationships are inspired by covariate models often used in PKPD models (Mould & Upton, 2012; Chung et al., 2021).

We choose the following parameters to get trajectories of different and realistic shapes.

$$\begin{aligned} g_0 &= 2.0 \\ d_0 &= 180 \\ \phi_0 &= 10 \end{aligned} \tag{11}$$

To generate the data, we create 2000 samples. For each sample, we draw static features from uniform distributions described below.

- $age \sim Uniform(20, 80)$
- $weight \sim Uniform(40, 100)$
- $initial\_tumor\_volume \sim Uniform(0.1, 0.5)$
- $dosage \sim Uniform(0.0, 1.0)$

For each sample, we evaluate the tumor volume function $y$ at 20 equally spaced time points on $[0, 1]$.

**Sine.** This synthetic dataset was created to show how our model can be used when we have one static feature that nevertheless renders trajectories with different trends. We use the following equation:

$$y(t) = \sin\left(\frac{2t\pi}{x}\right) \tag{12}$$

In this equation, $x$ is the static covariate. As we show in Figure 1, it can generate trajectories with different compositions. To create the dataset we generate 200 values for $x$ uniformly on $[0, 2.5]$, and for each $x$ we generate 20 uniform time points from 0 to 1 and evaluate $y$ on them.

**Beta.** This synthetic dataset was created to show how our model can be used when we have two static features that render trajectories with different trends. As the probability density functions of the beta

distribution are known for their variety of shapes, we decided to use them in our experiments. In particular, we define our trajectory by the following equation

$$y(t) = \frac{1}{\mathcal{B}(\alpha, \beta)} t^{\alpha-1} (1-t)^{\beta-1} \tag{13}$$

where $\mathcal{B}$ is the beta function. We treat $\alpha$ and $\beta$ as static features. To create the datasets, we create a grid of $(\alpha, \beta)$ pairs where $\alpha$ and $\beta$ are created by taking 30 values uniformly spread over $[1.0, 4.0]$. This gives us 900 samples in total. For each sample, we generate 20 uniform measurements on $[0, 1]$.

### D.2 BASELINES

**Linear-T.** We implement the linear model with elastic net regularization (Zou & Hastie, 2005) using `scikit-learn` package. We adapt them for time series forecasting by passing time as a feature. We perform hyperparameter tuning using Optuna (Akiba et al., 2019) and run it for 100 trials. We describe the hyperparameters we tune and their ranges in Table 7

Table 7: Linear-T hyperparameter ranges

| Hyperparameter | Considered values |
| --- | --- |
| alpha | Float from $[0.0, 1.0]$ |
| l1_ratio | Float from $[0.0, 1.0]$ |

**DecisionTree-T.** We implement decision tree regressor using `scikit-learn` package. We adapt them for time series forecasting by passing time as a feature. To keep the decision trees transparent, we restrict their depth to 5. We perform hyperparameter tuning using Optuna (Akiba et al., 2019) and run it for 100 trials. We describe the hyperparameters we tune and their ranges in Table 8

Table 8: DecisionTree-T hyperparameter ranges

| Hyperparameter | Considered values |
| --- | --- |
| max_depth | 5 |
| min_samples_split | Integers from $[2, 32]$ |
| min_samples_leaf | Integers from $[1, 32]$ |
| criterion | {squared error, friedman mse, absolute error} |

**GAM-T.** We implement GAM using the Explainable Boosting Machines (EBMs) (Lou et al., 2012) available in the `InterpretML` package (Nori et al., 2019). We adapt them for time series forecasting by passing time as a feature. By doing so, GAM finds a "baseline" trajectory which is the shape function associated with the $t$ variable. Then every prediction consists of the same baseline trajectory but shifted vertically depending on the values of other shape functions. Such model is very transparent but it is likely to underperform in settings where the trajectories have different trends for different features. We perform hyperparameter tuning using Optuna (Akiba et al., 2019) and run it for 100 trials. We describe the hyperparameters we tune and their ranges in Table 9

Table 9: GAM-T hyperparameter ranges

| Hyperparameter | Considered values |
| --- | --- |
| max_bins | Integers from $[128, 512]$ |
| outer_bags | Integers from $[4, 16]$ |
| inner_bags | Integers from $[0, 8]$ |
| learning_rate | Float (log) from $[0.001, 0.1]$ |
| max_leaves | Integer from $[1, 6]$ |
| min_samples_leaf | Integers from $[1, 4]$ |

**SINDy.** SINDYy (Brunton et al., 2016) is an ODE discovery algorithm that produces closed-form ODEs. We use the implementation in PySINDy library (de Silva et al., 2020; Kaptanoglu et al., 2022). We adapt SINDy to work with static features by treating static features as a constant control input. We perform hyperparameter tuning using Optuna (Akiba et al., 2019) and run it for 100 trials. We tune the optimizer_threshold by considering values from 0.001 to 0.1 (on a logarithmic scale), and we tune the derivative estimation method. We consider three different kinds of differentiation algorithms: finite difference, spline, and trend filtered. We show their corresponding hyperparameter ranges in Table 10

Table 10: SINDy differentiation algorithms: hyperparameter ranges

| Differentiation method | Hyperparameter | Considered values |
|---|---|---|
| Finite difference | $k$ | Integer from $[1, 5]$ |
| Spline | $s$ | Float (log) from $[0.001, 1]$ |
| Trend filtered | order | Integer from $[0, 2]$ |
| Trend filtered | $\alpha$ | Float (log) from $[0.0001, 1]$ |

**PySR.** PySR (Cranmer, 2020) is a symbolic regression library that uses genetic programming. Genetic programming algorithms highly depend on the time they are allowed to run for (as they can explore more possibilities). We put a time constraint for PySR that is equal to the time it took TIMEVIEW to run, including the hyperparameter tuning. Thus we do not perform hyperparameter tuning for PySR (we run with default parameters). Instead, after each run we look at the best equations of each length and we try each of them on the validation set. We then report the loss of the best-found equation on the test set. We choose the maximum length of the expression to be a maximum of 20 and the number of features times 3. We choose 20 as this is a reasonable length of expression to comprehend, and we choose 3 times the number of features as that is the length of a GAM that is usually considered interpretable. The parameters of PySR are listed in Table 11

Table 11: PySR hyperparameters

| Hyperparameter | Value |
|---|---|
| Binary operations | $+, -, \times, \div$ |
| Unary operators | $\log, \exp, \sin$ |
| maxsize | $\max(20, 3M)$ |

**RNN.** As a Recurrent Neural Network baseline, we use a multi-layer long short-term memory (LSTM) RNN as implemented in `pytorch`. We implement the encoder as multi-layer neural networks. We also include dropout (Srivastava et al., 2014) and batch normalization (Ioffe & Szegedy, 2015). The sizes of the hidden layers, activation functions, and dropout probability are fine-tuned. We perform hyperparameter tuning using Optuna (Akiba et al., 2019) and run it for 100 trials. We describe the hyperparameters we tune and their ranges in Table 12.

Table 12: RNN hyperparameter ranges

| Hyperparameter | Considered values |
|---|---|
| Activation | {ReLU,Sigmoid,Tanh,Leaky ReLU,ELU,SeLU} |
| Encoder hidden sizes | 3 integers from $[16, 128]$ |
| Encoder dropout probability | Float from $[0.0, 0.5]$ |
| Decoder dropout probability | Float from $[0.0, 0.5]$ |
| Decoder number of layers | Integer from $[1, 3]$ |
| Decoder hidden sizes | Integer from $[16, 128]$ |
| Learning rate | Float (log) from $[0.0001, 0.01]$ |
| Batch size | {64,128} |
| Weight decay | Float (log) from $[0.000001, 0.001]$ |

$\Delta t$-**RNN.** We adapt RNN described above to a setting with irregular measurements by passing the difference between time points as an additional feature (Yıldız et al., 2021).

**XGB-T.** We implement XGBoost (Chen & Guestrin, 2016) using `py-xgboost` package. We adapt it for time series forecasting by passing time as a feature. We perform hyperparameter tuning using Optuna (Akiba et al., 2019) and run it for 100 trials. We describe the hyperparameters we tune and their ranges in Table 13

Table 13: XGB-T hyperparameter ranges

| Hyperparameter | Considered values |
|---|---|
| n_estimators | Integers from $[10, 1000]$ |
| eta | Float (log) from $[0.001, 0.1]$ |
| min_child_weight | Integer from $[1, 10]$ |
| max_depth | Integer from $[3, 9]$ |
| gamma | Float (log) from $[0.00000001, 1.0]$ |
| subsample | Float from $[0.1, 1.0]$ |
| colsample_bytree | Float from $[0.1, 1.0]$ |
| lambda | Float (log) from $[1e - 8, 1.0]$ |

**CatBoost-T.** We implement CatBoost (Prokhorenkova et al., 2018) using `catboost` package. We adapt it for time series forecasting by passing time as a feature. We perform hyperparameter tuning using Optuna (Akiba et al., 2019) and run it for 100 trials. We describe the hyperparameters we tune and their ranges in Table 14

Table 14: CatBoost-T hyperparameter ranges

| Hyperparameter | Considered values |
|---|---|
| n_estimators | Integers from $[500, 2000]$ |
| learning rate | Float (log) from $[0.001, 1.0]$ |
| depth | Integer from $[1, 12]$ |
| boosting type | {Ordered, Plain} |
| bootstrap type | {Bayesian, Bernoulli, MVS} |
| l2_leaf_reg | Float (log) from $[0.01, 10]$ |
| early stopping rounds | Integers from $[10, 50]$ |

**LGBM-T.** We implement LightGBM (Ke et al., 2017) using `lightgbm` package. We adapt it for time series forecasting by passing time as a feature. We perform hyperparameter tuning using Optuna (Akiba et al., 2019) and run it for 100 trials. We describe the hyperparameters we tune and their ranges in Table 15

Table 15: LGBM-T hyperparameter ranges

| Hyperparameter | Considered values |
|---|---|
| n_estimators | $\{10, 50, 100, 200, 500, 1000, 2000, 5000\}$ |
| learning rate | Float (log) from $[0.00001, 10.0]$ |
| num_leaves | Integer from $[2, 256]$ |
| min_child_samples | Integer from $[1, 100]$ |
| reg_alpha | Float (log) from $[1e - 7, 10.0]$ or 0.0 |
| reg_lambda | Float (log) from $[1e - 7, 10.0]$ or 0.0 |

### D.2.1 SENSITIVITY ANALYSIS

In this section, we investigate how sensitive TIMEVIEW is to the number of basis functions $B$. We perform our experiments for different numbers of basis functions (from 5 to 16) on two real

datasets—Airfoil and Tacrolimus. The results can be seen in Figure 6. We observe that TIMEVIEW is not very sensitive to the number of basis functions and achieves good performance even when the number of basis functions is very low. That means it can have a good predictive power even if the compositions of the predicted trajectories are very short—and thus easy to understand.

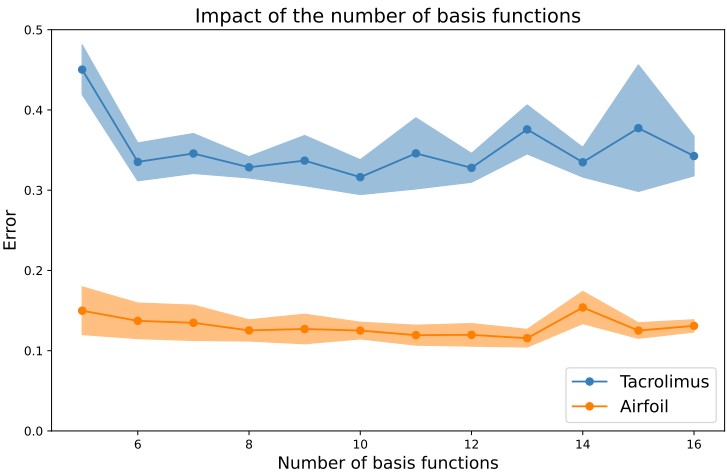

Figure 6: This figure shows how the error changes with the number of basis functions

## D.3 COMPUTATION TIME

It is important to note that training TIMEVIEW is basically the same as training a standard neural network for multi-output regression. Thanks to the preprocessing of the B-Spline basis functions there is no significant overhead due to time series forecasting. In fact, TIMEVIEW uniquely requires a single pass through the network to predict the whole trajectory no matter the time horizon. This may make it faster than methods that forecast trajectory by generating outputs sequentially. The visualization tool is applied only after the model is trained and works in real-time. The computation times (in seconds) can be seen in the Table 16 below. The run times are for all experiments for a given dataset. That includes 100 runs of hyperparameter tuning and 10 runs to get standard errors.

The experiments were performed on 12th Gen Intel(R) Core i7-12700H with 64 GB of RAM and NVIDIA GeForce RTX 3050 Ti Laptop GPU as well as on the 10th Gen Intel Core i9-10980XE with 60 GB of RAM and NVIDIA RTX A4000. All experiments take around 23.6h to run.

Table 16: Computation times (in seconds)

| method | Airfoil | Beta | Sine | Stress-Strain | Tacrolimus | Tumor | flchain |
|---|---|---|---|---|---|---|---|
| RNN | - | 3098 | 1121 | - | - | 4299 | 1034 |
| $\Delta t$-RNN | 691 | 2952 | 1265 | 1265 | 175 | 3901 | 1112 |
| XGB-T | 34 | 134 | 98 | 124 | 32 | 244 | 79 |
| LGBM-T | 25 | 364 | 71 | 172 | 9 | 93 | 30 |
| CatBoost-T | 70 | 1426 | 1098 | 882 | 93 | 1289 | 715 |
| PySR | 214 | 2018 | 1040 | 308 | 272 | 2977 | 1211 |
| SINDy | 64 | 292 | 77 | 2757 | 232 | 1136 | 35247 |
| GAM-T | 12 | 28 | 14 | 109 | 56 | 194 | 1139 |
| Linear-T | 1 | 2 | 1 | 2 | 1 | 4 | 5 |
| DecisionTree-T | 1 | 17 | 2 | 50 | 1 | 104 | 58 |
| TIMEVIEW | 191 | 1908 | 1022 | 194 | 135 | 2740 | 987 |

## D.4 LICENSES

The licenses of the software used in this work are presented in Table 17

Table 17: Software used and their licenses

| Software | License |
|---|---|
| gplearn | BSD 3-Clause "New" or "Revised" License |
| scikit-learn | BSD 3-Clause "New" or "Revised" License |
| numpy | liberal BSD license |
| pandas | BSD 3-Clause "New" or "Revised" License |
| scipy | liberal BSD license |
| python | Zero-Clause BSD license |
| PySR | Apache License 2.0 |
| interpret | MIT License |
| py-xgboost | Apache License 2.0 |
| pytorch | BSD-3 |
| pytorch lightning | Apache License 2.0 |
| tensorboard | Apache License 2.0 |
| matplotlib | Matplotlib license - BSD compatible |
| jupyter notebook | BSD 3-Clause "New" or "Revised" License |
| scikit-survival | GPL-3.0 license |
| pysindy | MIT License |
| catboost | Apache License 2.0 |
| lightgbm | MIT License |

# E ADDITIONAL DISCUSSION

## E.1 ANSWERING QUESTIONS ABOUT TIMEVIEW: TUMOR EXAMPLE

In this section, we demonstrate how we can use the visualization tool to understand TIMEVIEW predictions. In particular, we show how our tool allows us to answer the questions described in Section 1. We train TIMEVIEW on the synthetic tumor dataset (details in Appendix D).

**Q: Would the predicted tumor volume keep decreasing if we adjusted the treatment?**

This is a question on **level 1**—about the trend of the trajectory. Figure 7 shows that the current trajectory is decreasing. The colorful band under the `dosage` slider shows the compositions for different drug dosages. We can see that the trajectory is increasing for very low dosages (orange). For higher dosage (blue), the trajectory is first decreasing and then increasing. Finally, higher dosage (green and pink) corresponds to decreasing trajectories. To answer the question, it is enough to check at which points the bands change color. If the drug's dose is kept above 0.46, the trajectory will keep decreasing. The secondary plot in the right panel shows how the $y$-coordinate of the first transition point changes as the initial tumor volume changes. As expected, the plotted function is close to the identity. It is a good sanity check to confirm that our model learned that one of the static features corresponds to the initial measurement.

**Q: What feature changes would lower the minimum tumor volume?**

This is a question on **level 2**—about the property of a particular trend. Let us consider a lower drug dosage where the trajectory decreases and then increases (Figure 8). We observe that the minimum of the tumor corresponds to the $y$-coordinate of the second transition point. We can use the tool on the right to plot how the minimum depends on the dosage (or any other feature). We can see that the minimum volume decreases with an increased drug dosage.

**Q: How feature changes would impact the time this minimum is achieved?**

Similarly, we can consider the time when this minimum is achieved. We can easily do that by just switching from the $y$-coordinate of the transition point to its $t$-coordinate (Figure 9). We can see how the time depends on the dose of the drug. Crucially, we can see how by lowering the dose of the drug, we bring this time closer to 0 where it "merges" with the first transition point, and the composition of the trajectory changes. The trajectory becomes increasing (Figure 10).

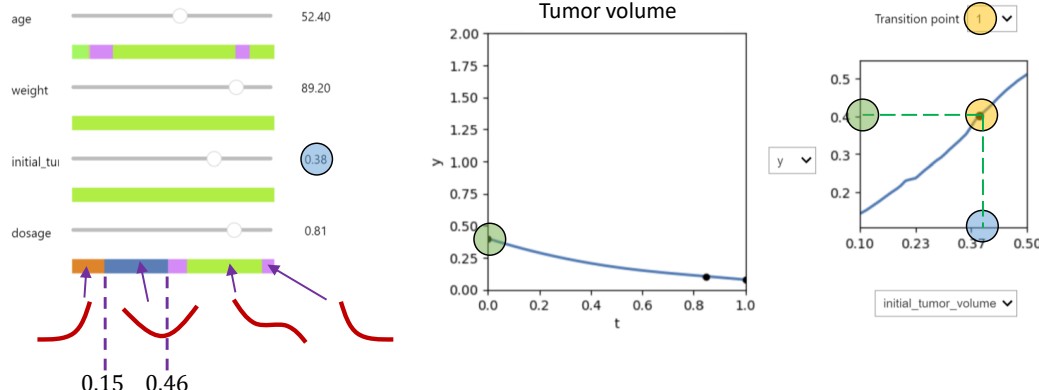

Figure 7: TIMEVIEW applied to the tumor dataset. Different dosages of the drug correspond to trajectories with different compositions. We can infer that if the drug's dose is kept above 0.46, the trajectory will keep decreasing. The right panel demonstrates that TIMEVIEW has learned that one of the static features corresponds to the initial measurement.

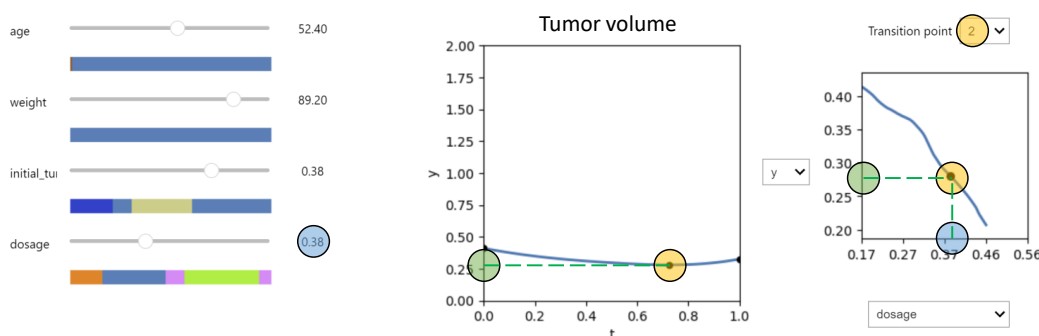

Figure 8: TIMEVIEW applied to the tumor dataset. The right panel shows how the trajectory's minimum depends on the drug's dosage.

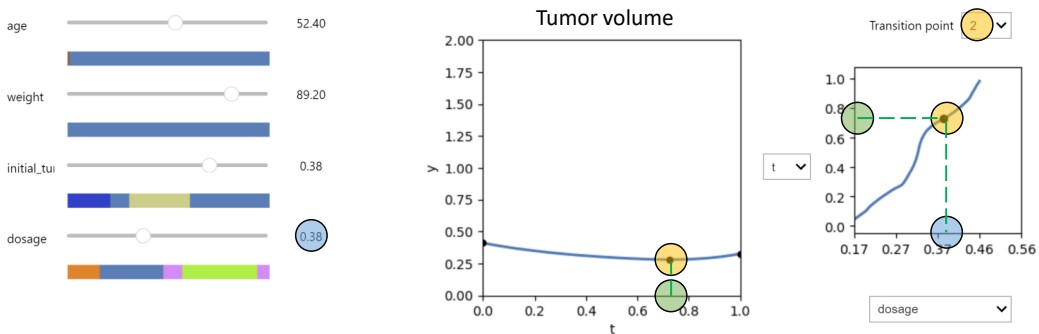

Figure 9: TIMEVIEW applied to the tumor dataset. The right panel shows how the time to achieve the minimum volume depends on the drug's dosage.

## E.2 CLOSED-FORM EXPRESSIONS

Whether a closed-form expression guarantees bi-level transparency or not is highly dependent on the exact for of the expression. For instance, a trajectory described by $y = \sin(2\pi x t)$ is transparent. We can understand the trend of the trajectory (the characteristic sinusoidal shape) and we can understand the impact of the covariate $(x)$ on the property of this trend (its frequency). However, some even very

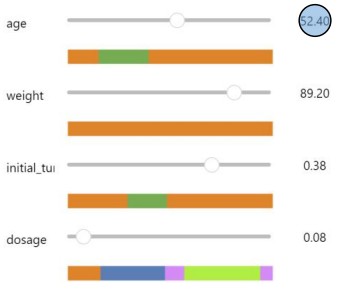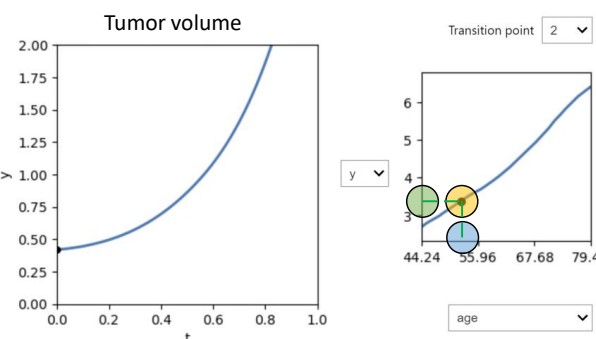

Figure 10: TIMEVIEW applied to the tumor dataset. For such low dose of the drug, the trajectory keeps increasing exponentially. We can also see that, for instance, the trajectory has the same trend regardless of the weight of the patient. The right panel shows how the final tumor volume depends on age.

simple expressions do not have this property. A good example is the expression for the probability density function of the beta distribution used in Section 7, i.e., $y(t) = \frac{1}{\mathcal{B}(\alpha,\beta)} t^{\alpha-1}(1-t)^{\beta-1}$, where $\mathcal{B}$ is the beta function, Even though the expression is very compact it is very challenging to infer the trend of the trajectory based on the features $\alpha$ and $\beta$. It is even harder to understand the impact of the features on the properties of these trends. That is why, we believe, the Beta dataset is a very interesting example for TIMEVIEW.

### E.3 CONTRAST WITH FEATURE IMPORTANCE METHODS

As we explain throughout the paper, current feature importance techniques are *bottom-up*, i.e., they calculate the importance scores with respect to individual prediction horizons. Below we show the results of applying SHAP (Lundberg & Lee, 2017) on top of LightGBM (adapted for time series forecasting) to the Sine (Figure 11), Beta (Figure 12), and Tacrolimus (Figure 13) datasets.

SHAP values assign an importance score to each feature. Features with positive scores impact the prediction positively, while those with negative SHAP values impact the prediction negatively. As we are interested in time series forecasting, the output of the algorithm is not a single outcome, but rather a whole trajectory. That is why we apply SHAP separately to a set of predictions at discrete time points.

While SHAP offers insights into the importance of specific features at discrete time points, TIMEVIEW takes this analysis a step further by providing a comprehensive understanding of feature impacts on the entire trajectory. It shows exactly how the prediction would change if the value of the feature is changed (i.e., how the trend of the trajectory and its properties would change).

Moreover, SHAP indicates feature importance at pre-specified time steps that may lack broader significance. In contrast, TIMEVIEW, captures the essence of the trajectory's behavior by demonstrating the impact on critical properties such as local maxima, minima, and inflection points even as their time coordinates vary.

Even in complex, high-dimensional scenarios, TIMEVIEW remains adept at addressing counterfactual queries. For instance, it can address the question of how the maximum of a trajectory changes if one of the variables is adjusted. This is of critical importance in pharmacology, where the trajectory is the drug concentration curve and the variable of interest is the drug dose. It also provides a clear understanding of how changes to the feature vector influence the trajectory's direction, for instance, transitioning it from ascending to descending. It is impossible to answer such queries using SHAP values.

In the table below we summarize examples of questions that can be asked about the model and whether they can be answered by SHAP and TIMEVIEW. We also observe that SHAP can always be used on top of TIMEVIEW to get additional insights. However, we cannot get TIMEVIEW-type explanations from just any black box model.

Table 18: Questions about a model and whether they can be answered using SHAP and TIMEVIEW

| Question | SHAP | TIMEVIEW |
|---|---|---|
| Which features are important? | Yes | Yes |
| Which features are important for the prediction at $t = 1.5$? | Yes | No |
| How does the dose of the drug impact the maximum concentration? | No | Yes |
| How does the feature vector need to be altered to change the trajectory from increasing to decreasing | No | Yes |

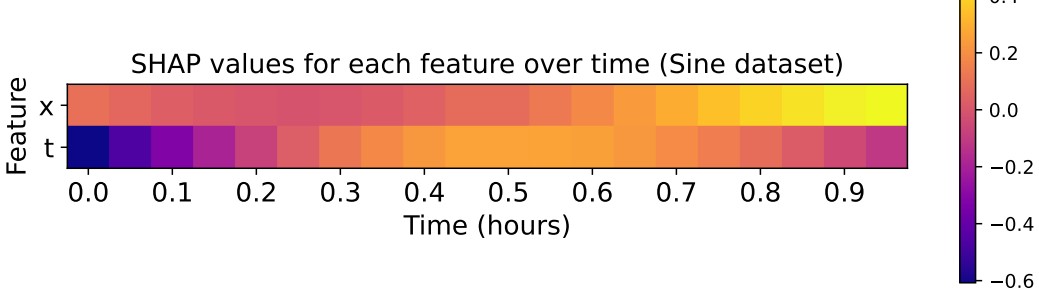

Figure 11: SHAP values for each feature over time calculated for LightGBM trained on the Sine dataset

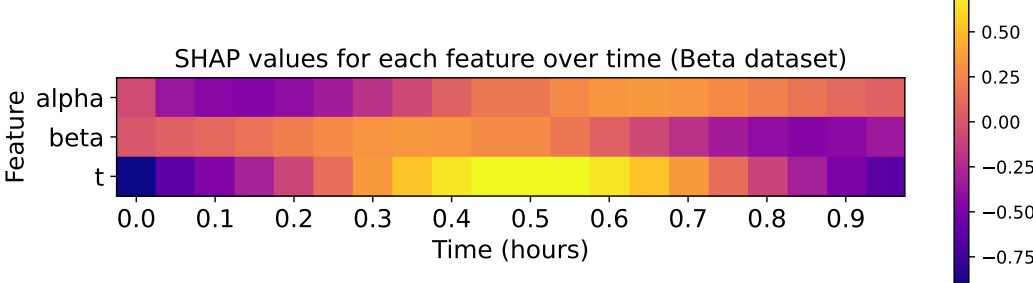

Figure 12: SHAP values for each feature over time calculated for LightGBM trained on the Beta dataset

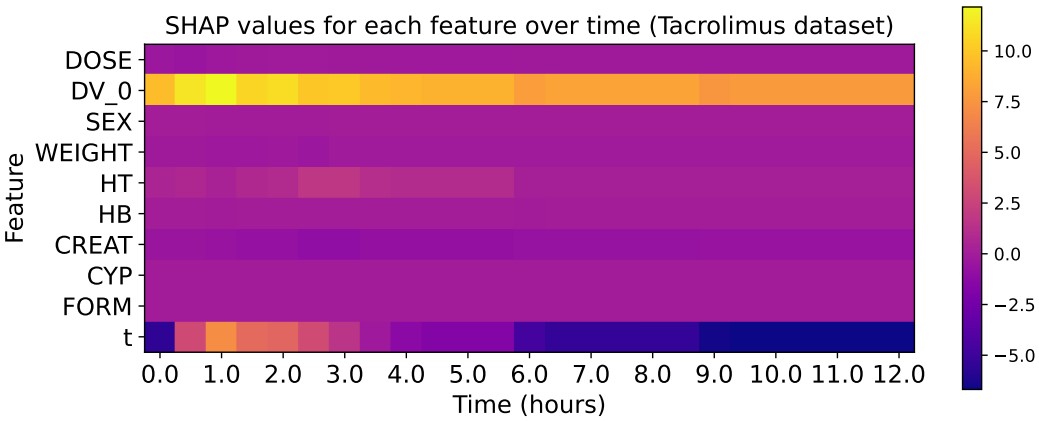

Figure 13: SHAP values for each feature over time calculated for LightGBM trained on the Tacrolimus dataset

### E.4 ADDITIONAL SNAPSHOTS OF THE INTERFACE

**Tacrolimus.** We apply our model to the Tacrolimus dataset (Woillard et al., 2011) that contains measurements of drug concentration in blood, and we present a snapshot of the trained model in Figure 14. The band under each slider shows the values of the composition map when we change the value of the corresponding slider but all other features are fixed. This allows us to answer questions such as "*What If*", "*How to be that*", and "*How to still be this*" from the XAI Question Bank (Liao et al., 2020) discussed in Section 1. For instance, in Figure 14, we choose the second transition point that corresponds to the maximum concentration of the drug. On the secondary plot, on the right, we can see how the position of this maximum changes as we vary any of the covariates. Currently shown plot indeed shows that if the dose of the drug is decreased then the maximum concentration decreases as well. We can also see that it is not an entirely linear relationship.

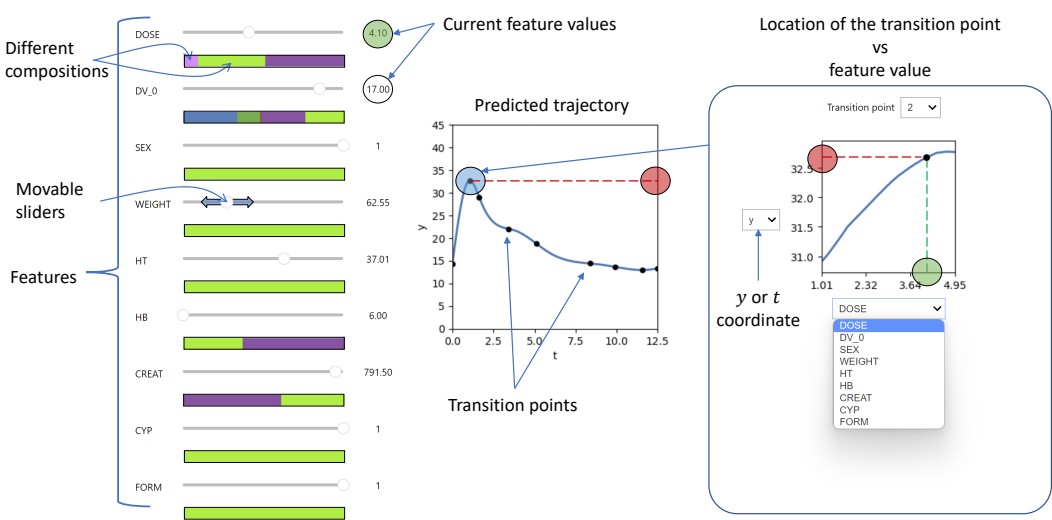

Figure 14: This figure shows the full interface where we are interested in perturbing one feature at a time (thus we use colorful bands instead of contour plots)

**Sine.** We also show a snapshot of our interface when TIMEVIEW is applied to the Sine dataset— Figure 15.

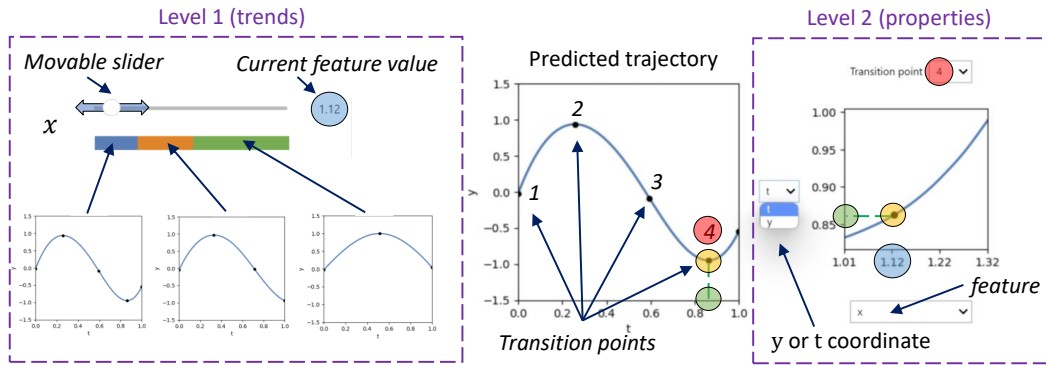

Figure 15: Snapshot of our dynamical visualization of TIMEVIEW. Our model adheres to *bi-level* transparency—a *top-down* approach that focuses on the trend of the trajectory (Level 1) and the properties of the particular trend, e.g., transition points (Level 2). The left panel shows how the trajectory trend changes when one of the features is perturbed. The right panel investigates the position ($y$-coordinate) of the second transition point (local minimum) as the initial tumor volume changes.

**Beta.** Sometimes, we may be interested in the effect of perturbing two features at the same time. We visualize it using a colorful 2D contour plot. Figure 16 shows TIMEVIEW applied to the Beta dataset.

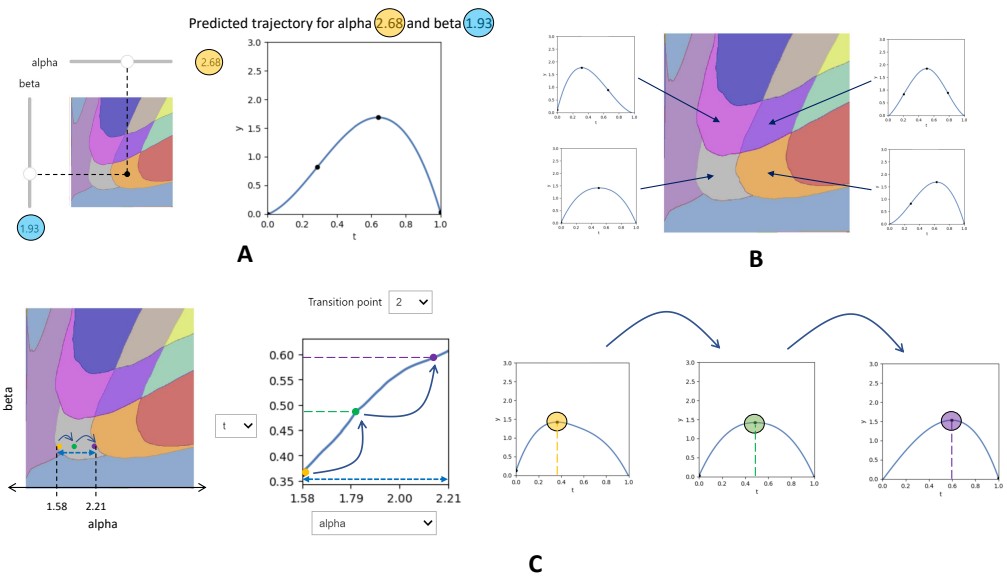

Figure 16: This figure shows how our interface can be used to understand perturbations of two features at the same time. Panel **A** shows how you can move the sliders to move around the feature space and update the predicted trajectory on the plot on the right. Panel **B** shows different trends corresponding to different regions of the feature space. Panel **C** shows how we can choose any of the transition points and see how their position changes as we change any of the features. In particular, we show how the second transition point, the maximum of the trajectory, moves to the left as the $\alpha$ increases.

## E.5 Insight Into Results

**Some methods perform better on real datasets than synthetic ones.** As we explain in Section 7, we create the two synthetic datasets (Sine and Beta) to be challenging in the following way: the trend of the trajectory changes significantly for different features. The same is true for the Tumor datasets. The tumor growth model we use was proposed to fit many different cancer trajectories (Wilkerson et al., 2017). That includes exponential increase, exponential decay, decrease and then increase. Although real data can be noisy, the bigger variability of trends makes it harder for some methods.

**GAM-T struggles on the Tumor dataset.** As we mention in Section 6, GAM-T's predicted trajectories are all parallel. They are equal to the shape function $f_t(t)$ shifted vertically by $f_1(x_1) + \ldots, + f_M(x_M)$ for static features $x$. That means it can model only one trend. As the focus of the synthetic datasets was on different trends, it is not surprising that GAM-T performs very poorly on them. In particular, the Tumor dataset contains trajectories of different trends (e.g., exponential increase, exponential decay, decrease and then increase).

## E.6 Limitations And Future Works

**Static inputs.** TIMEVIEW is a particular application of bi-level transparency for time series forecasting from static features. We hope future works will extend it to settings where the input may contain the previous part of the trajectory or other exogenous time series.

**Modelling the trajectory as a cubic spline.** We chose to model the trajectory as a cubic spline because cubic splines are flexible, and this representation allows for a straightforward composition extraction (see Section 5 and Appendix C). However, not all trajectories can be effectively approximated by a cubic spline. Future works should investigate other ways to represent the trajectory that would still make it amenable to composition extraction.

**Regularization.** Although we use the $L2$ penalty to constrain how quickly the compositions change as we change the input features, future research could explore constraining the learned composition map for improved interpretability or designing better penalties for overly long and abruptly changing compositions.

**Likelihood of a sample.** The visualization interface allows for showing all combinations of input variables, even ones that do not appear in reality. In the future, the interface can be enhanced to display information about the likelihood of a particular sample (or a whole set of samples).

**Confidence bounds.** We could apply a technique like Deep Ensembles to get uncertainty estimates on the coefficients of the B-Spline basis functions which we could then transform into uncertainty estimated for the whole trajectory. Another interesting approach would be to adapt uncertainty estimation to our conceptual framework of trends and properties, i.e., how uncertain we are about a particular trend.

**Modeling seasonality.** Our framework could accommodate seasonality by modeling seasonal changes as a sine function where the neural network determines the frequency, magnitude, and offset (the same way as the B-Spline coefficients). Our model would be defined as:

$$g(\boldsymbol{x})(t) = \sum_{k=1}^{B} h(\boldsymbol{x})_b \phi_b(t) + A(\boldsymbol{x}) \sin(\omega(\boldsymbol{x})t + \psi(\boldsymbol{x})) \tag{14}$$

where $A : \mathbb{R}^M \to \mathbb{R}$, $\omega : \mathbb{R}^M \to \mathbb{R}$, $\psi : \mathbb{R}^M \to \mathbb{R}$ are either separate neural networks or they share weights with the encoder $\mathbf{h}$.

Then, we can treat the compositions in the same way as the current implementation of TIMEVIEW but with three additional properties to monitor (in addition to transition points). This allows us to answer questions like "How would the variability increase if I decrease this particular feature?".

**Alternatives to cubic splines.** As we describe in Section 5, we chose cubic splines because we needed a space of trajectories that satisfies the following two criteria.

- The set of dynamical motifs $\mathcal{S}$ is compatible with the class of predicted trajectories $\hat{\mathcal{Y}}$.
- For every $\hat{y} \in \hat{\mathcal{Y}}$ we can calculate its composition $\mathcal{C}_{\mathcal{S}}[\hat{y}]$

We could use many other basis functions, such as a sine basis or some types of wavelets. They could satisfy the first condition, but the composition would likely need to be calculated numerically and not analytically as it is done now. In case we are interested in a different set of motifs (please see Appendix A.4 for examples), other choices may also be possible. For instance, for range motifs or monotonic motifs, Haar wavelets may be a good choice, and we suspect the compositions can be calculated efficiently.

**Applications to traditional XAI techniques for static predictions.** Adhering to our framework provides additional output next to the standard forecasted trajectory: the current composition and the coordinates of the transition points. Traditional XAI techniques for regression and classification can be applied to these additional outputs instead of individual trajectory points to gain more meaningful explanations. Thus, techniques such as dimensionality reduction, feature importance methods, local surrogates, and counterfactual explanations can now be extended to time series forecasting. This opens up numerous potential extensions and applications of our approach.

### E.7 INHERENTLY DISCRETE TRAJECTORIES

Modeling a phenomenon as a continuous system is an established practice in sciences and engineering. That includes settings mentioned in our paper, such as disease modeling or drug concentration in blood. Of course, in practice, we only observe discrete measurements, which may be irregular and noisy. However, sometimes the phenomenon cannot be modeled as a continuous system. This happens where the set of time points is inherently discrete, and there is no notion of time "in between the time points". For instance, the number of sunshine hours for each day would be an inherently discrete trajectory as there is no meaningful "number of sunshine hours" between two consecutive days. We note that our conceptual work in Section 2 still accommodates such scenarios. However, the formalism in Section 4 must be adapted.

### E.8 TRAJECTORY CHANGES

In Section 2, we claim that although a prediction of a regression algorithm can change in only 3 ways (increase, decrease, remain constant), a trajectory can change in numerous ways. To see that, let us consider a set of time points $\mathcal{T} = \{1, 2, \ldots, 9, 10\}$. The function can increase, decrease, or remain constant for each time step. Thus, the function described by these ten values can change in $3^{10} = 59049$ ways (which is much bigger than 3—the number of ways an output of a regression model can change). We describe this in the paragraph on "increased cognitive load" in Section 2. It is infeasible for a human to reason about all these different possibilities, which motivates our proposal of a top-down approach. The situation becomes even more complex where the trajectory is not defined over ten time points but over a whole real interval—which is the focus of our work.

## F EXTENDED RELATED WORKS

**Transparent models for static predictions.** Standard transparent methods for static predictions include linear/logistic regression, scoring systems (Ustun & Rudin, 2016), decision trees/rule lists (Angelino et al., 2018; Hu et al., 2019), and generalized additive models (GAMs) (Hastie & Tibshirani, 1986; Lou et al., 2012). Such methods can often be used for time series forecasting by passing the time $t$ as an additional feature. They often satisfy bi-level transparency but have poor performance. For instance, all trajectories predicted by linear regression and GAMs are parallel; thus, they cannot model different trends (Section 7). Decision Trees capture non-additive interactions, enabling flexible forecasting models. However, they require many splits to approximate the ground truth, leading to poor performance or incomprehensibility (Section 7).

**Closed-form expressions.** Closed-form expressions are mathematical formulas composed of a finite number of variables, binary operators $(+, -, \times, \div)$, well-known functions (e.g., $\sin, \exp, \log$), and constants. For instance, $\sin(x^2) - e^{2.1y}$. A machine learning area that aims to find such expressions fitting the data is called Symbolic Regression (La Cava et al., 2021). This area originated from attempts to unearth equations describing physical systems through machine learning (Schmidt & Lipson, 2009; Udrescu & Tegmark, 2020; Holt et al., 2023; Biggio et al., 2021; D'Ascoli et al., 2022). Differential equations represent another category of mathematical expressions that draw significant interest in the scientific community. Numerous algorithms have been proposed for discovering Ordinary Differential Equations (ODEs) (Brunton et al., 2016; Qian et al., 2022; Kaheman et al., 2020; Messenger & Bortz, 2021a) and Partial Differential Equations (Rudy et al., 2017; Long et al., 2019; Raissi & Karniadakis, 2018; Messenger & Bortz, 2021b; Kacprzyk et al., 2023). Although some mathematical expressions may satisfy bi-level transparency, this is not guaranteed as it depends on the actual form of the found equation. In fact, reparametrization of equations so that their parameters reflect quantities of key theoretical interest is an active area of research (Preacher & Hancock, 2015). We delve into this subject in greater detail and provide examples in Appendix E.

**Overview of XAI techniques.** While our research focuses on transparent models, the landscape of post-hoc explainability methods has experienced significant growth. Such methods are used to explain the predictions of a black box model. These include feature importance methods (Ribeiro et al., 2016; Lundberg & Lee, 2017) (also called saliency methods) that highlight which features the model is sensitive to, example importance methods (Ghorbani & Zou, 2019; Pruthi et al., 2020; Crabbe et al., 2021) that identify important training samples, and concept-based explanations (Kim et al., 2018). Relatively little attention has been devoted to time series (Barredo Arrieta et al., 2020), but a few recent methods aim to extend feature importance to this setting (Crabbé & Schaar, 2021; Leung et al., 2023).

**Feature importance for time series.** While our research focuses on transparent models, many saliency (or feature importance) methods have been developed to highlight which features the model is sensitive to (Ribeiro et al., 2016; Lundberg & Lee, 2017). Although these methods have been extended to time series *inputs* (Crabbé & Schaar, 2021; Leung et al., 2023), limited work has been done to extend them specifically to time series *outputs*. Current XAI techniques either assume the output is a scalar (Siddiqui et al., 2019) (e.g., time series classification (Hao & Cao, 2020)), treat the trajectory as a single object (Gao et al., 2023)—thus do not show *how* a feature changes the trajectory—or show a saliency map at each predicted point separately (Pan et al., 2020), thus allowing only for a bottom-up understanding of the predicted trajectory. Saliency methods can be broadly divided into Gradient-based (Sundararajan et al., 2017; Shrikumar et al., 2017), Perturbation-based

(Zeiler & Fergus, 2014), and Attention-based methods (Vaswani et al., 2017; Alaa & van der Schaar, 2019; Lim et al., 2021). Other important examples include SHAP (Lundberg & Lee, 2017) based on Shapley values and LIME (Ribeiro et al., 2016) that fits a local linear model.

**Shapelets and motifs.** As our method discusses the shape of the trajectory, it may seem related to shapelet-based methods (Ye & Keogh, 2009). However, these methods are usually used for data mining and classification tasks. They aim to find subsequences of a time series that represent the most important patterns of each class and thus can be used to distinguish between them (Chen et al., 2022). Similarly, motif discovery identifies short repeating patterns in the time series (Torkamani & Lohweg, 2017) usually for insights into the problem or classification tasks.

**Statistical methods.** Although methods like ARIMA or Hidden Markov Models (Rabiner & Juang, 1986) have potential for compact state transition equations, these models generally fail to meet bi-level transparency standards due to the complexity of understanding how the input feature changes influence the entire trajectory—they do not allow for top-down understanding as the trajectory is constructed sequentially.

