# OpenReview forum: "Towards Transparent Time Series Forecasting"
_ICLR.cc/2024/Conference — ICLR 2024 poster_

### Official Review · Reviewer_pJsr · 2023-10-26

**Soundness:** 2 fair
**Presentation:** 2 fair
**Contribution:** 2 fair
**Rating:** 6
**Confidence:** 3

**Summary:**

The authors introduce TIMEVIEW, a top-down transparent framework designed for trajectory forecasting with static variables. They aim for two levels of transparency, explaining the forecasting through both its trends and properties. To formalize this approach, the authors introduce the concepts of motifs and compositions, avoiding the tracking of individual trajectory values. TIMEVIEW comprises two components: a predictive model based on B-Spline basis functions, and an algorithm for calculating the composition map to facilitate easy visualization. The efficacy of TIMEVIEW is assessed using four real-world datasets and three synthetic ones, and it is compared against several competitors. Transparency is demonstrated through a user-selected example on a synthetic dataset, with additional examples provided in the appendices.

**Strengths:**

- The focus on transparency in forecasting is extremely important.
- The paper is generally well-written and interesting.
- The proposed framework appears sound and potentially useful in specific domains.

**Weaknesses:**

- The limitation to static variables is a significant drawback. Additionally, the model can visualize the effects of only two variables, which is another strong limitation.
- The paper should provide a more robust assessment of the model's transparency, rather than relegating this to the appendix.
- No specific code references were found.

**Questions:**

- The focus on static variables seems to limit the framework's utility in true time-series forecasting. Given that you claim to forecast time-dependent variables based solely on fixed attributes, isn't this more akin to regression analysis? This approach seems niche.
- I order to avoid the two variables limit, have you considered visualizing the contour plot using techniques like t-SNE, PCA, UMAP, or similar?
- Since transparency is the primary goal of your paper, more space should be devoted to proving this claim. For instance, I had difficulty understanding Figure 4.
- Considering Table 2, which classifies types of motifs, have you evaluated how performance would change if motifs were characterized as sets of classes rather than as continuous values? If the end user ultimately requires such a table, perhaps classification would suffice.
- In the related works, you distinguish your method from shapelet-based methods, stating that these are primarily used for data mining and classification tasks. However, if these shapelet methods are unsupervised (e.g., Karlsson, Isak, Panagiotis Papapetrou, and Henrik Boström. "Generalized random shapelet forests." _Data mining and knowledge discovery_ 30 (2016): 1053-1085.), they can also be applied to regression—and consequently, forecasting through reduction.

---

> ### Author Response · Authors · 2023-11-15
> **Dear Reviewer pJsr [1/2]**
>
> Dear Reviewer pJsr,
>
> Thank you for the time and attention you spent reviewing our work. We are encouraged by your kind words about an important problem we tackle, the presentation of our work, and the applications of our approach. Your insights are invaluable in helping us refine our work and better communicate its contributions.
>
> We structure our response into four sections:
> 1. Presentation
> 2. Clarifications
> 3. Limitations
> 4. Extensions
>
> # 1. Presentation
> We appreciate your comments about the presentation of our work, as they allow us to improve the manuscript further. In particular, we acknowledge your comment about moving some of the transparency assessment from the appendix into the main text. We have updated Figure 1 to show a snapshot of our interface applied to the tumor volume problem. We hope that this clarifies the advantage of our framework compared to other approaches and provides a more coherent narrative.
>
> We acknowledge your comment regarding Figure 4. We have now added additional snapshots of the interface to Appendix E.4 which should help in understanding how our model can visualize the effect of perturbing two features at the same time.
>
> **Actions taken**
> - Exchanged Figure 1 for a different snapshot of the interface.
> - Added additional snapshots of the interface to Appendix E.4.
> # 2. Clarifications
> ## TIMEVIEW works for more than two dimensions
> We would like to thank you for your point about two dimensions. This allows us to clarify this point in the paper. TIMEVIEW can indeed show the effects of more than two variables. This can be seen on many snapshots of the interface (e.g., Figures 7-10 in Appendix E.1 and Figure 14 in Appendix E.4). We have replaced Figure 1 with a snapshot demonstrating this capability more clearly.
>
> As we focus on how our interface can be used to answer specific questions (in particular, "What if", "How to be that", and "How to still be this" from the XAI question bank (Liao et al., 2020)) we consider scenarios when we are interested in understanding the change in the prediction as we change one or two of the covariates *at a time* (while all the other features are kept fixed). We note that, usually, people are interested in perturbing one or two features *at a time* as understanding change in more than two features would be cognitively very challenging. For instance, "Would the tumor keep decreasing if we adjusted the dose of the drug?" requires understanding the change in one dimension.
> ## Code
> As we say in the reproducibility statement,
> > The code to reproduce the results and for the visualization tool can be found in supplementary materials.
> ## Terminology
> As there are many conflicting terminologies used throughout the literature, we specify the exact setting in Section 2.1, where we contrast time series forecasting with regression and classification models. We further describe the specific problem formulation in Section 3. We identified one place where we used the term "time series setting" and updated it to "time series forecasting" to make it consistent with the rest of the paper.
>
> ## Using shapelets for forecasting
> Although shapelets are becoming increasingly useful for many tasks, we are unsure how they can be currently leveraged for time series forecasting—for predicting a whole trajectory rather than a single label. They are usually used as a feature extractor (e.g., for time series classification), whereas we are interested in predicting a whole function. We envision that, in future works, shapelets may become a part of a larger pipeline where interpretable features are extracted from additional time series input and passed to TIMEVIEW as static features. This would extend TIMEVIEW to more complex input types.
>
> **Actions taken**
> - Updated "time series setting" to "time series forecasting" for consistency.
>
> `rebuttal continues in the next comment...`

---

> > ### Author Response · Authors · 2023-11-15
> > **Dear Reviewer pJsr [2/2]**
> >
> > `...rebuttal continued`
> >
> > # 3. Limitations
> > ## Clear scope of the work
> > We want to highlight that we clearly describe the scope of our work (Section 3, Section 8, Appendix E.6). Although the conceptual framework in Section 2 is agnostic to the type of input, we focus on settings with static features to develop formalism and demonstrate a practical implementation.
> > ## Reasoning behind the scope
> > The main goal of our work is to extend the notion of transparency to a novel class of outputs (trajectories), going beyond standard regression and classification algorithms (see Section 2 for more details). Thus, it provides a concrete answer to the question: What does it mean to understand the change in the output in time series forecasting? Of course, to ensure transparency, we also need to understand changes in the input. To provide a clear exposition, we focus on settings where this is straightforward, and we hope future work will extend it to more complex input types. We believe there is sufficient novelty without this extension.
> > ## Real-life settings with static features
> > As we describe in Section 3, there are numerous scenarios that include predicting a trajectory from static features. Many of them include applications to medicine and pharmacology. Static features include the baseline patient's covariates (such as age, weight, or genetic information) and the information about the type of treatment or the dose of the drug. The trajectory of interest can be a tumor volume over time or drug plasma concentration. If necessary, one or a few initial observations at pre-specified times can also be considered to be static features. More examples of such scenarios can be found in finance (predicting stock values from the company's static data), time-to-event problems (predicting the survival or the hazard function), or physics (modeling some dynamical systems from their initial conditions). In some scientific or engineering domains, time can be even replaced by other continuous variables. For instance, when modeling stress-strain or current-voltage curves.
> > # 4. Extensions
> > ## Dimensionality reductions techniques
> > Although dimensionality reduction is not necessary for answering the questions we consider in our work, it undoubtedly can provide additional information about the model. This is exactly what we describe in "Applications" in Section 8. Adhering to our framework provides additional output next to the standard forecasted trajectory: the current composition and the coordinates of the transition points. Traditional XAI techniques for regression and classification can be applied to these additional outputs instead of individual trajectory points to gain more meaningful explanations. Thus, techniques such as dimensionality reduction, feature importance methods, local surrogates, and counterfactual explanations can now be extended to time series forecasting. This opens up numerous potential extensions and applications of our approach. We have added an additional section in Appendix E to discuss these kinds of applications.
> >
> > **Actions taken**
> > - Added a section to Appendix E discussing the potential applications and extensions of our framework
> >
> >
> > Thank you for your questions and suggestions. They will undoubtedly improve the quality of our paper. Please let us know if you have any further questions or comments. We are more than happy to address them.
> >
> > **References**
> > Liao, Q. V., Gruen, D. & Miller, S. Questioning the AI: Informing Design Practices for Explainable AI User Experiences. in _Proceedings of the 2020 CHI Conference on Human Factors in Computing Systems_ 1–15 (Association for Computing Machinery, 2020).

---

### Official Review · Reviewer_PiQk · 2023-10-30

**Soundness:** 3 good
**Presentation:** 3 good
**Contribution:** 2 fair
**Rating:** 6
**Confidence:** 3

**Summary:**

This article discusses the importance of transparent machine learning models, particularly in high-stakes domains like healthcare, finance, and criminal justice. While transparent models have been proposed for classification and regression, time series forecasting poses unique challenges for transparency.

The article introduces a top-down framework for bi-level transparency in time series forecasting, aiming to understand both higher-level trends and lower-level properties of predicted time series. It emphasizes the need to comprehend changes in trajectory, which is more complex than single-label outputs.

The article contrasts this with the challenges of using a more traditional bottom-up approach that focuses on individual time points and values, which may not provide a holistic understanding of the entire time series.

To implement the proposed framework the article introduces TIMEVIEW, a transparent machine learning model for time series forecasting based on static features. The model uses B-Spline basis functions and an encoder to match feature vectors to trajectory descriptions.

**Strengths:**

- The problem addressed in the article is important and under-developed in the existing literature.
- The presentation is clear and intuitive.
- The idea of a top-down framework in this context appears novel.

**Weaknesses:**

- The main contribution in the method appears to be in suggesting a pre-determined dimension reduction to the time series and operating in that new space.
- The framework appears limiting in applications where time series variability is an important aspect of the interpretation.

**Questions:**

- How would your framework behave when one of the features influences the variability of the time series?
- Are there alternatives to the cubic spline setup when using your framework?
- Are there ways to incorporate domain knowledge/priors into this framework?
- Does your framework seamlessly apply to the common time series setting of a single long sample, i.e. when predicting on overlapping chunks of the time series with a corresponding panel of features? Would there be a concern with the selection of nodes being non-congruent for the overlapping samples?

---

> ### Author Response · Authors · 2023-11-15
> **Dear Reviewer PiQk [1/2]**
>
> Dear Reviewer PiQk,
>
> Firstly, we express our deep gratitude for your thoughtful and constructive feedback and for recognizing our work's importance, clarity, and novelty. Your insights inspired us to consider new perspectives and potential improvements. We truly value your contributions as they enhance the quality and impact of our research. Below, we address your comments and questions, incorporating your valuable suggestions.
>
> We structure our response into the following three sections.
> 1. Contributions
> 2. Clarifications
> 3. Extensions
>
> # 1. Contributions
> We appreciate your comments about our contributions as they allow us to further explain our work's main contributions. In particular, our contributions can be divided into four parts.
> - (a) conceptual
> - (b) theoretical
> - (c) methodological
> - (d) practical
> ## (a) conceptual
> Current XAI approaches for time series forecasting explain individual time points (bottom-up). That may be sufficient in some settings, but often, we are interested in a more holistic understanding of the whole predicted trajectory (top-down). For instance, current approaches can answer the question, "How would the predicted tumor volume at $t=1.5$ change if we adjusted the treatment?". While somewhat informative, often it is much more meaningful to answer questions such as "Would the predicted tumor volume keep decreasing if we adjusted the treatment?" (this is a question on level 1—about the trend of the trajectory) or "What feature changes would lower the minimum tumor volume?" (this a question on level 2—about the property of a particular trend). This approach allows us to introduce bi-level transparency that provides a more systematic and precise definition of a transparent forecasting model.
> ## (b) theoretical
> Based on the conceptual framework developed in Section 2, we introduce a mathematical formalism that makes the concepts rigorous. This formal treatment and the theoretical results are necessary for the method we develop in Section 5. We aimed to make the formalism as general as possible so that many future methods can be developed for different choices of motifs (please look at Appendix A.4, where we discuss different examples of motifs).
> ## (c) methodological
> TIMEVIEW consists of two parts: the predictive model and an algorithm for calculating the composition map. Both are driven by the conceptual and mathematical framework described earlier and constitute their particular instantiations. We note that the model's architecture is chosen precisely to ensure that the set of dynamical motifs is compatible with the class of the predicted trajectories and that the compositions can be calculated analytically and efficiently. We also ensured that much of the computation could be performed before the training. Thus, the actual training loop is no more complex than training a standard feed-forward neural network for multioutput regression, and the composition extraction can be performed in real-time. In addition, TIMEVIEW can work with observed trajectories containing different numbers of irregular measurements. While we chose to focus on prediction from static features, the conceptual framework is agnostic to the type of inputs.
> ## (d) practical
> We developed an interface that allows us to visualize the composition map and answer various questions about the model, both about the composition of the trajectory (level 1) and the coordinates of the transition points (level 2). Moreover, we show that our method performs much better than other transparent algorithms and achieves a comparable performance to black-box models.
> # 2. Clarifications
> We assume that the setting you describe refers to a dataset where different samples are observed at different time intervals. For instance, $\textbf{x}_1$ is observed only on $[0,1.5]$ and sample  $\textbf{x}_2$ is observed only on $[1,2]$, etc. We want to clarify that TIMEVIEW applies to such settings as long as we can assume all samples have their underlying trajectories defined over some long interval (e.g., $[0,3]$) and are just measured in parts of it. The knots are global—the same for all predicted trajectories—so we do not observe an issue of incongruent knots. Theoretically, it does not matter whether the trajectories are measured at the same intervals or are regularly sampled (please see the loss function in Equation 4). In practice, it may be more difficult to correctly approximate the parts of the trajectory where the data is sparse.
>
> `rebuttal continues in the next comment...`

---

> > ### Author Response · Authors · 2023-11-15
> > **Dear Reviewer PiQk [2/2]**
> >
> > `...rebuttal continued`
> >
> > # 3. Extensions
> > ## Variability
> > TIMEVIEW does not model the noise introduced by the measurement process. It assumes that there is a true underlying trajectory $y_*$ that we want to approximate (see Section 3). However, if the variability you describe refers to a deterministic property of the predicted trajectory, for instance, seasonality, then this can be accommodated by our framework! We could model seasonal changes as sine function where the frequency, magnitude, and offset are determined by the neural network (the same way as the B-Spline coefficients). Our model would be defined as:
> > $$ g(\mathbf{x})(t) = \sum_{k=1}^B h(\mathbf{x})_b \phi_b(t) + A(\mathbf{x}) \sin(\omega(\mathbf{x}) t + \psi(\mathbf{x}))$$
> > where $A:\mathbb{R}^M\rightarrow \mathbb{R}$, $\omega:\mathbb{R}^M\rightarrow \mathbb{R}$, $\psi:\mathbb{R}^M\rightarrow \mathbb{R}$  are either separate neural networks or they share weights with the encoder $\mathbf{h}$.
> >
> > Then, we can treat the compositions in the same way as the current implementation of TIMEVIEW but with three additional properties to monitor (in addition to transition points). This allows us to answer questions like "How would the variability increase if I decrease this particular feature?". We have now added a discussion of this extension to Appendix E.
> > ## Alternatives to cubic splines
> > As we describe in Section 5, we chose cubic splines because we needed a space of trajectories that satisfies these two criteria:
> > > 1. The set of dynamical motifs $\mathcal{S}$ is compatible with the class of predicted trajectories $\hat{\mathcal{Y}}$.
> > > 2. For every $\hat{y} \in \hat{\mathcal{Y}}$ we can calculate its composition $\mathcal{C}_{\mathcal{S}}[\hat{y}]$
> >
> > We described above how cubic splines can be extended by adding a sine function. We could use many other basis functions, such as a sine basis or some types of wavelets. They could satisfy the first condition, but the composition would likely need to be calculated numerically and not analytically as it is done now.
> >
> > In case you are interested in a different set of motifs (please see Appendix A.4 for examples), other choices may also be possible. For instance, for range motifs or monotonic motifs, Haar wavelets may be a good choice, and we suspect the compositions can be calculated efficiently. We have now included a discussion about alternatives to cubic splines in Appendix E.
> > ## Domain knowledge/priors
> > One way to incorporate priors is to choose the number of knots for the cubic splines. The bigger the number of knots, the more complex trajectories can be fit. More generally, as mentioned above, different basis functions (possibly for different motifs) can be chosen to approximate the trajectory.
> >
> > **Actions taken**
> > - Added discussion to Appendix E about an extension that includes modeling variability
> > - Added discussion to Appendix E on alternatives to cubic splines
> >
> > Your feedback has been instrumental in elevating the quality of our work. We are grateful for your engagement and hope we have addressed all your concerns. Please let us know if you have any further questions. We are more than happy to address them.

---

> > > ### Comment · Reviewer_pJsr · 2023-11-22
> > >
> > > Thank you for your responses and clarifications.

---

> > > > ### Author Response · Authors · 2023-11-22
> > > > **Thank you**
> > > >
> > > > Thank you again for your constructive feedback. We hope to have addressed all the concerns you raised in your review, which we believe have enhanced the clarity and rigor of our submission.
> > > >
> > > > If you find that our responses and improvements have sufficiently addressed your concerns, we would be grateful if you would consider revising your score accordingly.
> > > >
> > > > Many thanks,
> > > >
> > > > The Authors

---

> > > ### Comment · Reviewer_PiQk · 2023-11-22
> > >
> > > Thank you for your clarifications and improvements. I will maintain my score.

---

### Official Review · Reviewer_eJfZ · 2023-10-30

**Soundness:** 3 good
**Presentation:** 3 good
**Contribution:** 3 good
**Rating:** 8
**Confidence:** 3

**Summary:**

Interpretability in time series forecasting is typically regarded a post-hoc task, and takes a "bottom-up" approach that relies heavily on the observations at individual timestamps. In this paper, the authors argue that this view is overly granular and non-intuitive, whereas focusing on the trends and properties of the time series trajectory would be more informative. In the spirit of explainable ML, the authors then propose the notion of bi-level transparency, along with a framework (TIMEVIEW) that encodes time series are a composition of B-splines. The interpretability aspect is further complemented by the development of a UI tool that allows for further explainability, counterfactual reasoning and sensitivity analysis on the time series. The paper concludes with a comparison to other forecasting methods, where the authors note that predictive performance remains comparable with the aded benefit of enhanced interpretability.

**Strengths:**

- The paper is well-written, and a pleasure read. It is well structured, and the included visualisations successfully convey the paper’s propositions. I commend the authors for their attention and care to detail when preparing this submission.
- I appreciated how the authors considered different flavours of interpretability here, as illustrated in the Figure on Pg 2. The complete example presented in Appendix E also conveys the paper’s contributions very effectively.
- The inclusion of a notebook for executing the aforementioned analysis also serves as a useful guide for practitioners looking to experiment with this framework, and lays the foundations for future extensions of this work.

**Weaknesses:**

- Although the authors already include a few examples of real-world problems where the tooling can be applied, I think that a more fleshed out experiment that runs throughout the whole paper can be especially helpful for conveying the contributions in a clearer manner (Appendix E accomplishes this very well).
- The comparison to other methods towards the end of the paper addresses a question I had regarding the trade-offs between interpretability and performance, but this constraint could also be featured more prominently throughout the paper, such that the proposed methodology is consistently presented in light of this compromise.
- I would be interested in learning more about instances where the proposed simulation is *not* suitable, and what properties of time series one would need to inspect before deciding to apply this technique (beyond the requirement for static features).

**Questions:**

See sections above.

** Post-rebuttal Update **

Increased score from 6 to 8.

---

> ### Author Response · Authors · 2023-11-15
> **Dear Reviewer eJfZ**
>
> Dear Reviewer eJfZ,
>
> Thank you very much for your thoughtful and constructive feedback on our paper. We are delighted to hear that you found the paper well-written and the visualizations effective. Your insights have been invaluable in helping us refine our work.
>
> We structured our response into two sections.
> 1. Presentation
> 2. Limitations
>
> # Presentation
> ## Experiment that runs throughout the whole paper
> We agree with your suggestion regarding the integration of a continuous example throughout the paper. While the detailed discussion in Appendix E.1 exemplifies our contributions effectively, we understand the impact of a coherent narrative in the main text. To address this, we have further developed the tumor volume example introduced in the beginning. We have integrated this theme more consistently across the paper, replacing Figure 1 with a more illustrative snapshot of the tumor volume interface. We have also ensured that this example is referenced in Section 2, further tying together the paper's various sections. Please let us know if you find that these revisions improves the clarity and exposition.
> ## Trade-off between interpretability and performance
> Thank you for your suggestion about a more prominent exposition of the trade-off between interpretability and performance. To address this, we have expanded our discussion on this aspect, now featuring it more prominently in the introduction (Contributions) and Section 5 (Implementation).
>
> **Actions taken**
> - Exchanged Figure 1 for a different snapshot of the interface.
> - Referred to the tumor example in Section 2.
> - Added discussion about the trade-off between interpretability and performance in the introduction and Section 5.
>
> # Limitations
> We have already touched upon it in Appendix E.6 (Limitations and future works). Although it is not a limitation of our conceptual framework, TIMEVIEW may struggle to predict trajectories that cannot be well approximated using a cubic spline. For instance, curves with sharp changes in the derivative or discontinuities. This could be addressed by considering motifs that are not dependent on any smoothness constraints (for instance, monotonic motifs as in Example 3 in Appendix A.4) and by using a different space of predicted trajectories $\hat{\mathcal{Y}}$.
>
> We hope this response adequately addresses your concerns. Should you have any additional questions or require further clarification, we are more than willing to engage in further discussion. Thank you again for your valuable feedback and the opportunity to improve our work.

---

> > ### Comment · Reviewer_eJfZ · 2023-11-22
> > **Acknowledgement of Rebuttal**
> >
> > Thank you for detailed responses across all reviews. I still consider the paper to be well-written, and an interesting contribution to the literature. Crucially, the paper is also in a fairly "complete" stage and fit for publication.
> > I am raising my score to an 8 to reflect this.

---

> ### Author Response · Authors · 2023-11-22
> **Thank you!**
>
> Dear Reviewer eJfZ,
>
> Thank you very much for your positive and constructive feedback during the rebuttal period. We are deeply grateful for your recognition of our paper as a well-written and valuable contribution to the field. We appreciate your decision to increase your score.
>
> Kind regards,
>
> Authors

---

### Official Review · Reviewer_nunX · 2023-11-01

**Soundness:** 2 fair
**Presentation:** 1 poor
**Contribution:** 2 fair
**Rating:** 3
**Confidence:** 2

**Summary:**

In this paper, Authors propose a transparent model for Time Series Forecasting, which should provide direct explanation for the results generated by the model. They investigate a top-down approach that decompose a time series in trend and properties.

They demonstrate the efficiency of their platform TIMEVIEW with time series based on static characteristics. The proposal enables to explore the impact of varying the static characteristics onto the output, while demonstrating reasonable performances.

**Strengths:**

Lots of effort are made to describe the context, the goal and the importance of transparency.

The baselines models are well-presented, and all parameters are clearly defined, which will greatly help researchers reproduce these results.

**Weaknesses:**

In my opinion, the writing style is not adapted and will confuse or have readers get lost.
There are too many references to future sections/definitions/details, which complexify the reading. For instance, Authors introduce some concept, but do not give further explanation (even simplified one), but rather referred to specific sections. Which makes readers go back and forth through the paper.

In addition, there is no conclusion to this paper, which should give readers the most important points to remember from this study.

It is surprising that in a paper oriented in TSF there is only one TSF baseline (RNN).

Finally, I think that there are too many claims that are not referenced or back-up in this paper, which diminishes its credibility.

(Note the introduction and discussion style are really unconventional for this type of paper)

**Questions:**

I had difficulties to fully understand the paper as I had to go back and forth during the reading. And I have a Time Series Forecasting (TSF) background and the I think the terminology used in the paper got me confused. However, the most important question I have is:

Where is the explainability? If I understand correctly, the proposal can help determine the impact of varying given static parameters (figure 4) on the output of the model. However, it does not infer on why it has this specific impact (the hidden relation in the model). Current, AI models are considered as black boxes because what is happening inside the model to link input and output is complex and too difficult for human to understand. But in my opinion the proposal is not different from such models. It does not give clear relation between input characteristics that generate a given output.  I am not an expert on XAI, but in my opinion, it looks like more to an exploratory platform than a XAI model.
Furthermore, why is it more comprehensible than other baselines? Someone could set up a platform to visualize the impact of changing the input of a linear model on the produce output, but will it be a good XAI model?

And here are some more comments that in my opinion would need to be treated to increase the current score.

# Bottom-up versus Top-down

The bottom-up approach mentioned by the Authors is rather a “bottom” approach in my understanding. Meaning that models used the raw data (usually regularly spaced) to do TSF. Authors need to clarify this point what is the “up” especially for reader to understand the difference with the proposed approach.
Is it the following:

*Bottom-up*: from time points predict time points and define a trajectory, but

> understanding trajectory from time points is not natural for human

Then, why not from time point determine the trend (for instance with moving average), min, max and other statistics (with sliding window) that will make understanding "simpler"? Why the top-down approach is better?

*Top-down*: Is this approach doing from time point define a trajectory then predict a trajectory for which we first look at trend and properties and then time points? Or from time point predict trajectory for which we first look at trend and properties and then time points?

In addition, Authors never demonstrated the issue with the bottom-up approach and why do bottom-up approaches are not bi-level transparent. In related work, they say,

# Terminology
## Feature
In the example box in the introduction, what is the definition of feature? And instance? These terminologies are not clear and may confuse readers.

In usual TSF, feature (or variate) are other time series that support the prediction of the target (what Authors also called exogenous features). But it looks like a feature in the introduction example is a characteristic or a specific time step of the time series.  Authors would need to clarify these points to avoid confusion when reading.

In addition, does instance here mean an input sample of the model?

## Regression versus Time series
In my opinion, __time series setting__ is not an appropriate terminology.
it should be multistep forecasting or something similar.

## Knots
Definition of knots should not be in appendix.


# Continuous time series

> interval [0, T] ⊂ R, where T ∈ R is a time horizon, and the underlying trajectory is continuous

Every time series are discrete as values are points/snapshot taken at a given time. Authors might review this sentence or at least clarify their point.

> Thus, understanding a (continuous) trajectory by individual values is unnatural for humans

But again, time series are never continuous. Even with a very low frequency sampling, it is still a set of points that, in the best scenario, can be matched or approximated by a function/trajectory.

# Function versus trajectory

> there are numerous ways a function can change

I disagree for each time steps the function will either increase, decrease or remain constant, same as regression. Probably, Authors want to say that the function(trajectory?) on the prediction window, may have different shapes depending on different outputs.

# Claims

> often satisfy bi-level transparency but have poor performance

But where is the proof for that claim? Which model from the experiments are bottom-up approaches?

> However, these methods are not directly applicable to time series forecasting

It should be backed up by some reference. Linear Regression can be applicable to time series forecasting. Indeed, Linear models such as proposed in [1] have proven to be competitive with latest Transformer-based models, which indeed are less explainable.

# Comparison with decomposition proposal or multi-resolution

[1] and [2] are proposal to decompose time series in trend and seasonality (with either transformer or linear based), why such method would be less explainable than the proposal?

[1] https://arxiv.org/pdf/2205.13504.pdf

[2] https://arxiv.org/pdf/2106.13008.pdf

What are the differences of the proposal with multi-resolution solution such as [3] (learning from different temporal granularity)?

[3] https://link.springer.com/chapter/10.1007/978-3-031-05933-9_6



# Paper needs a proof-reading
 * ”different prediction” wrong opening brackets
 * Smoothing parameter is s and motif is also s change to avoid confusion

---

> ### Author Response · Authors · 2023-11-15
> **Dear Reviewer nunX [1/3]**
>
> Dear Reviewer nunX,
>
> Thank you for dedicating your time and expertise to review our paper thoroughly. We genuinely appreciate the detailed feedback. We have structured our response to address each point directly, reflecting our commitment to enhancing the clarity and quality of our manuscript. We structured our rebuttal into three sections:
> 1. Clarifications
> 2. Presentation
> 3. Comparisons
>
> In every section, we address all your comments point by point.
>
> # 1. Clarifications
> We are afraid that some parts of our paper were misunderstood. We appreciate your remarks as they allow us to improve the clarity of our paper further. We provide clarifications about the following.
> - 1.1 Bottom-up versus top-down
> - 1.2 Explainability
> - 1.3 Continuous time series
> - 1.4 Trajectory changes
> - 1.5 Claims
> ## 1.1 Bottom-up versus top-down
> We first want to clarify that "bottom-up" and "top-down" refers to how the trajectory is comprehended, not how the prediction is generated. We use these terms to differentiate between XAI techniques rather than contrast model architectures. Your questions made us realize that this distinction can be made even more explicit in our paper, so we added a footnote in Section 2 to prevent any future confusion.
>
> Current XAI methods usually produce explanations for individual time points ("bottom"). These, in turn, have to be pieced together by a human to understand changes in trends ("up") from changes at individual time steps. Thus we call it "bottom-up". This is challenging, and we discuss the limitations of this approach in Section 2.2 and then demonstrate them experimentally in Figures 11-13 in Appendix E.3. As we describe, these explanations do not allow for answering essential questions such as "Would the predicted tumor volume keep decreasing if we adjusted the treatment?". In contrast, our top-down approach (explained in Section 2.3), where we understand the trend of the trajectory ("top") and then its properties ("down"), allows for answering such questions. We show it in Appendix E.1, where we review several questions and show how our interface allows us to answer them.
>
> ## 1.2 Explainability
> One of the goals of eXplainable Artificial Intelligence (XAI) is to create models or methods that answer meaningful questions about the models and their predictions (Liao et al., 2020). One of the biggest advantages of our approach compared to other techniques is the ability to answer many (often more meaningful) questions. Please look at Appendix E.1, where we demonstrate in detail how our model can be used to answer such questions, and Appendix E.3, where we demonstrate and discuss the limitations of current feature importance methods. In particular, all bottom-up approaches allow for answering questions about a particular time point. For instance, "How would the predicted tumor volume at $t=1.5$ change if we adjusted the treatment?". In specific scenarios, this may be sufficient. However, often it is much more meaningful to answer questions that require a different kind of interpretability. For instance, "Would the predicted tumor volume keep decreasing if we adjusted the treatment?" (this is a question on level 1—about the trend of the trajectory) or "What feature changes would lower the minimum tumor volume?" (this a question on level 2—about the property of a particular trend). We treat these exact questions in Appendix E.3. We have added explicit references to this section in the introduction and updated Figure 1, so that it corresponds to the questions about the tumor volume trajectory. We hope that by following through these multiple examples, a future reader will better grasp the significance of our approach.
>
> ## 1.3 Continuous time series
> Thank you for your question about continuous time series. It allows us to improve further our explanation of the setup, which is already included in the sentence directly following the one you quote:
> > Of course, in practice, we only observe discrete samples of y, which may be irregular and noisy.
>
> By providing a stronger link between these two sentences, we hope to prevent future confusion about this issue. Modeling a phenomenon as a continuous system is an established practice in sciences and engineering. That includes settings mentioned in our paper, such as disease modeling or drug concentration in blood. It is also well represented in the machine learning literature (Brunton et al., 2016; Chen et al., 2018). We have also added a discussion to Appendix E about scenarios where the phenomenon cannot be studied as a continuous system. However, our conceptual framework in Section 2 still applies even in these settings.
>
> `rebuttal continues in the next comment...`

---

> > ### Author Response · Authors · 2023-11-15
> > **Dear Reviewer nunX [2/3]**
> >
> > `...rebuttal continued`
> >
> > ## 1.4 Trajectory changes
> > In your comment, you raise the following interesting point:
> > > I disagree for each time steps the function will either increase, decrease or remain constant, same as regression.
> >
> > This is exactly what we refer to as a "bottom-up" understanding of the trajectory exercised by most of the XAI approaches, and we devote Section 2.2 to exploring its limitations. Based on your remark, we believe that a more in-depth discussion addressing your statement would significantly improve the accessibility of this section and improve the overall clarity. Thus, we have added a dedicated section to Appendix E and refer to it at the end of Section 2.1.
> >
> > Let us consider a set of time points $\mathcal{T} = \\{1,2,\ldots,9,10\\}$. As you state, the function can increase, decrease or remain constant for each time step. Thus, the function described by these ten values can change in $3^{10} = 59049$ ways (which is much bigger than $3$—the number of ways an output of a regression model can change). This is what we describe in the paragraph on "increased cognitive load" in Section 2.2. It is infeasible for a human to reason about all these different possibilities, which motivates our proposal of a top-down approach.
> >
> > ## 1.5 Claims
> > Thank you for your question about the proof of our claim that certain models
> > > often satisfy bi-level transparency but have poor performance.
> >
> > It allows us to better join this sentence with the two sentences directly following it that already provide explanations and point to experimental evidence.
> > > For instance, all trajectories predicted by linear regression and GAMs are parallel; thus, they cannot model different trends (Section 7). Decision Trees capture non-additive interactions, enabling flexible forecasting models. However, they require many splits to approximate the ground truth, leading to poor performance or incomprehensibility (Section 7).
> >
> > Similarly, we appreciate your question about evidence for our statement
> > > these methods are not directly applicable to time series forecasting—when we want to predict a whole trajectory rather than a single label.
> >
> > It allows us to connect it better with the sentence directly preceding it that explains that these models cannot be used *directly* for time series forecasting as, by definition, they issue static predictions (single labels), not whole trajectories.
> >
> > We hope these changes improve our sections' cohesiveness and make it easier to find corresponding evidence for our claims.
> >
> > **Actions taken**
> > - Added a footnote in Section 2 to clarify that top-down and bottom-up refer to trajectory comprehension, not how the trajectory is computed.
> > - Added explicit references to Appendix E.1.
> > - Exchanged Figure 1 for a different snapshot of the interface.
> > - Improved the discussion about continuous time series in Section 2
> > - Added a section to Appendix E about scenarios where the phenomenon cannot be studied as a continuous system.
> > - Added a section to Appendix E about trajectory changes and a reference to it in Section 2.
> > - Improved the links between sentences in Section 1 and Section 6
> >
> > # 2. Presentation
> > We welcome your comments on how the presentation of our work could be improved. Below, we address your comments about
> > - 2.1 Explanations
> > - 2.2 Conclusion
> > - 2.3 Terminology
> > - 2.4 Minor issues
> > ## 2.1 Explanations
> > We appreciate your remark about further explaining some of the terms when they are first introduced. We have now improved the description of *trends* and *properties* in the introduction, hopefully making the paper more straightforward.
> > ## 2.2 Conclusion
> > We have renamed our last "Discussion" section to "Discussion and Conclusion". It discusses the applications, limitations, and future works.
> > ## 2.3 Terminology
> > ### Features
> > The questions in the box in the introduction are only prototypical examples from question classes identified by Liao et al. (2020). Concrete questions are shown at the bottom of the box. We adjusted the terms in the box to avoid possible confusion. We specify the exact problem formulation in Section 3.
> > ### Time series forecasting
> > As there are many conflicting terminologies used throughout the literature, we specify the exact setting in Section 2.1, where we contrast time series forecasting with regression and classification models. We further describe the specific problem formulation in Section 3. We identified one place where we used the term "time series setting" and updated it to "time series forecasting" to make it consistent with the rest of the paper.
> > ### Knots
> > We have added a short explanation of knots in Section 5.
> > ## 2.4 Minor issues
> >
> > ### Quotation marks
> > We have now fixed some quotation marks that evaded our proofreading efforts. Thank you for finding them.
> >
> > `rebuttal continues in the next comment...`

---

> > > ### Author Response · Authors · 2023-11-15
> > > **Dear Reviewer nunX [3/3]**
> > >
> > > `...rebuttal continued`
> > >
> > > ### Smoothing parameter
> > > We note that the "smoothing parameter s" appears only in Appendix C.3, where we refer to a function `UnivariateSpline` from the `scipy` library, and we use the letter from its original documentation to denote its argument. Although "smoothing parameter s" is not used in any other place in the paper, we added a footnote in Appendix C.3 to avoid confusion. Table 4 in Appendix A.1 lists all essential symbols used in the paper.
> > >
> > > **Actions taken**
> > > - Improved introduction of trends and properties.
> > > - Renamed the "Discussion" section to "Discussion and Conclusion".
> > > - Adjusted terms in the example box in the introduction.
> > > - Updated "time series setting" to "time series forecasting" for consistency.
> > > - Added an explanation of knots in Section 5.
> > > - Fixed a few quotation marks.
> > > - Added a footnote in Appendix C.3.
> > >
> > > # 3. Comparison
> > > Thank you for your suggestions about additional comparisons. We want to clarify the difference between a forecasting model and how it used for explanations. It is easier for us to explain the difference between our approach and another forecasting model if it is clear where the explainability of the other model lies. Still, we take an educated guess. We divide our reply into two sections:
> > > - 3.1 Transformers
> > > - 3.2 Linear models
> > > ## 3.1 Transformers
> > > Transformers are sometimes treated as explainable models because their attention masks can be visualized for every time point predicted. Thus, as described in Section 6 and elaborated in Appendix F, these methods can be treated as feature importance (or saliency) methods for time series. Similarly to gradient-based or perturbation-based methods, they allow only a bottom-up understanding of the predicted trajectory. It is impossible to understand how the change in the output will influence the trend and properties of the predicted trajectory. Thus, the transformer models you suggested do not satisfy bi-level transparency. We appreciate their novel architecture that decomposes time series into a trend and seasonality, but it is not clear to us how that model structure aids in answering the meaningful questions described in our work.
> > > ## 3.2 Linear models
> > > We thank you for your comment
> > > > Linear Regression can be applicable to time series forecasting.
> > >
> > > as it allows us to clarify this earlier in the paper. Indeed, as we explain in Section 6 and later demonstrate in Section 7, linear regression can be used for time series forecasting, and it yields a model that is bi-level transparent but sadly performs poorly (Table 3). We have now included this point in Section 1 to foreshadow what is happening later in the paper. We hope that this will remove the unnecessary ambiguity.
> > >
> > > The particular linear model proposed by Zeng et al. (2023) that you refer to is a very simple model designed for more complex input data and high-dimensional trajectories. Applied to our setting, it can be considered a very special case of the GAM-T model (to which we already compared our method in Section 7) with the following differences.
> > > - All trajectories need to be measured at the same time points.
> > > - It requires one of the static features to be the initial condition $y_0$.
> > > - It ignores all other features as it "shares weights across different variates and does not model any spatial correlations".
> > > - Whereas GAM-T trajectories can be interpreted as being shifted vertically when one of the features is changed, the trajectories of this model can be interpreted as being multiplied by $\lambda$ if the initial condition $y_0$ is multiplied by $\lambda$.
> > >
> > > Assuming the type of explanations described above, this model would be considered bi-level transparent, but it would offer very poor performance.
> > >
> > > **Actions taken**
> > > - Included point about adapting models like linear regression for time series forecasting in the introduction.
> > >
> > >
> > > We believe the revisions made will substantially improve the paper and its comprehension. Please let us know if you have any further questions or comments. We are eager to address them.
> > >
> > > **References**
> > >
> > > Brunton, S. L., Proctor, J. L. & Kutz, J. N. Discovering governing equations from data by sparse identification of nonlinear dynamical systems. _Proc Natl Acad Sci USA_ **113**, 3932–3937 (2016).
> > >
> > > Chen, R. T. Q., Rubanova, Y., Bettencourt, J. & Duvenaud, D. K. Neural Ordinary Differential Equations. in _Advances in Neural Information Processing Systems_ vol. 31 (Curran Associates, Inc., 2018).
> > >
> > > Liao, Q. V., Gruen, D. & Miller, S. Questioning the AI: Informing Design Practices for Explainable AI User Experiences. in _Proceedings of the 2020 CHI Conference on Human Factors in Computing Systems_ 1–15 (Association for Computing Machinery, 2020).
> > >
> > > Zeng, A., Chen, M., Zhang, L. and Xu, Q. Are transformers effective for time series forecasting?. In _Proceedings of the AAAI conference on artificial intelligence_ Vol. 37 (2023)

---

> ### Comment · Reviewer_nunX · 2023-11-21
>
> I would like to thank the Authors for taking time in replying to my comments. However, I will maintain my current score as Authors did not manage to convince me with their answers.
>
> Please find below additional comments based on Authors' rebuttal.
>
> ## Bottom-up versus top-down
> > demonstrate them experimentally in Figures 11-13 in Appendix E.3
>
> Appendix says:
> > TIMEVIEW takes this analysis a step further by providing a comprehensive understanding of feature impacts on the entire trajectory [...]
>
> It might be only me, but I don't see why TIMEVIEW is more comprehensive. Authors would need to provide the TIMEVIEW impact feature for these datasets (Appendix E1 is only Tumor example) for the readers to appreciate this "comprehensive understanding". As mentioned in my original review, this is another claim without rigorous proof.
>
> ## Trajectory change
> > It is infeasible for a human to reason about all these different possibilities, which motivates our proposal of a top-down approach.
>
> Why would TIMEVIEW make things easier then? Every time a human modifies a value from the parameters the potential trajectory change, but human still has no clue on the reason for the impact.
>
> ## Claims
> >all trajectories predicted by linear regression and GAMs are parallel; [...]
>
> Where can Authors see that? Section 7 is only a table with average (standard deviation) MSE...
>
> ## Comparison with Transformers
> > It is impossible to understand how the change in the output will influence the trend and properties of the predicted trajectory.
>
> In my opinion, it is also impossible to understand how the change in the input (Authors typo?) will influence the trend and properties with TIMEVIEW. You can visually see the impact of changing parameters values, but not understand the reason for these changes.
>
> ## Comparison with Linear
> > but it would offer very poor performance.
>
> How can you affirm that it will offer very poor performance without trying it? There are dozens of papers showing that Linear-based models are effective for TSF, so why it would be different for your datasets? Or is it that your proposal applied to only specific type of dataset? If so, you should revise paper to make clear that this method cannot be applied to any TSF.

---

> > ### Author Response · Authors · 2023-11-22
> > **Dear Reviewer nunX [1/3]**
> >
> > Dear Reviewer nunX,
> >
> > We appreciate your continuous engagement with our work. Please find our response below.
> > # Bottom-up versus top-down
> >
> > The main advantage of TIMEVIEW compared to standard bottom-up approaches (and feature importance methods in particular) is that **it allows for answering more meaningful questions** about the model.
> >
> > Although current methods can answer questions about an impact of a feature on the prediction at a pre-specified time point (e.g., "How would the predicted tumor volume at $t=1.5$ change if we adjusted the treatment?"), these time points may lack broader significance. TIMEVIEW can demonstrate the impact on both the general shape of the trajectory and critical properties such as local maxima, minima, and inflection points (e.g., "Would the predicted tumor volume keep increasing if we adjusted the treatment? How would the minimum of the tumor volume trajectory change?"). Table 18 in Appendix E.3 already summarizes four canonical questions a person may ask about the model and whether SHAP and TIMEVIEW can answer them. We reproduce it below for your convenience.
> >
> > | Question                                                                                              | SHAP | TIMEVIEW |
> > | ----------------------------------------------------------------------------------------------------- | ---- | -------- |
> > | Which features are important?                                                                         | Yes  | Yes      |
> > | Which features are important for the prediction at $t = 1.5$?                                           | Yes  | No       |
> > | How does the dose of the drug impact the maximum concentration?                                       | No   | Yes      |
> > | How does the feature vector need to be altered to change the trajectory from increasing to decreasing? | No   | Yes      |
> >
> > **We respectfully disagree with the reviewer's claim that "this is another claim without rigorous proof"**. We have pointed in our rebuttal that evidence for our claims can be found in the following or preceding sentences. Moreover, we show how TIMEVIEW can answer questions discussed above in Appendix E.1. Please note that we show snapshots of our interface applied to four datasets overall (**not only Tumor**). These can be found in Figures 1, 4, 7, 8, 9, 10, 14, 15, 16. We contrasted TIMEVIEW with bottom-up methods in Appendix E.3 in Figures 11-13. We also included the interface as a jupyter notebook in the supplementary materials.
> > # Trajectory change
> > We recognize that your primary concern about our work is that TIMEVIEW does not allow us to "understand the reason" for the impact on the algorithm's output. We interpret your statement as referring to the fact that our method is not *simulatable*. Below, we describe what simulatability is, why it is challenging to define for time series forecasting, and why it may not be very useful in practice.
> > ## Simulatability
> > Although our framework allows for answering meaningful counterfactual questions, it does not show the exact process by which the values of the trajectory are calculated from the inputs. In other words, it does not satisfy the very strict requirement of simulatability (Lipton, 2018) described as "for a model to be fully understood, a human should be able to take the input data together with the parameters of the model and in reasonable time step through every calculation required to produce a prediction". Although useful for *static predictions*, directly applying this definition to time series forecasting is challenging. Does "produce a prediction" refer to evaluating the values of the trajectory at some pre-specified points? How many points? Which points? Note that this corresponds to a bottom-up approach to understanding a trajectory and, as explained earlier, may not be very useful in practice. Is the only way to be simulatable to have a closed-form mathematical expression (like in symbolic regression)? In particular, we want to highlight the following.
> > 1. The issue of simulatability for time series forecasting has yet to be investigated and needs to be defined. (Barredo Arrieta et al., 2020)
> > 2. Whether it is possible to have a flexible and simulatable model for time series forecasting is an open problem.
> > 3. As simulatability is closely linked to bottom-up understanding, it is not clear if it has clear advantages in practice (disadvantages of bottom-up understanding discussed in Section 2.2 and earlier in this response).
> >
> > Thus, we introduce a different notion of transparency and highlight its advantages in Section 2.3. Let us explain with a concrete example how TIMEVIEW makes understanding the change in the output easier than other bottom-up approaches.
> >
> > `reply continues in the next comment...`

---

> > > ### Author Response · Authors · 2023-11-22
> > > **Dear Reviewer nunX [2/3]**
> > >
> > > `... reply continued`
> > >
> > > ## TIMEVIEW makes understanding the change in the trajectory easier: a concrete example
> > >
> > > Let $y:[0,1] \rightarrow \mathbb{R}$ be a trajectory given by $y(t) = t^2$. Consider some change of the input such that the new trajectory is now described by $y(t)=-t^2$ . We contrast a bottom-up description with a top-down description.
> > > ### Bottom-up description
> > > Bottom-up approach focuses on the change at individual time points. That would amount to:
> > > - For $t=0.0$ the value stays the same;
> > > - for $t=0.2$ the value decreases by $0.08$;
> > > - for $t=0.4$ the value decreases by $0.32$;
> > > - for $t=0.6$; the value decreases by $0.72$;
> > > - for $t=0.8$ the value decreases by $1.28$;
> > > - for $t=1.0$ the value decreases by $2.0$.
> > >
> > > We claim that it is often not the most useful way to represent the change as it is difficult to comprehend all these individual changes to piece together a more holistic understanding. It is also restricted only to a few time points.
> > > ### Top-down description
> > > Using top-down approach, we would say: "The trajectory changes from increasing and convex starting at $0.0$ and ending at $1.0$ to a decreasing concave trajectory starting at $0.0$ and ending at $-1.0$". We argue that this description is easier to comprehend and already includes the essential aspects of this change.
> > >
> > > A similar example can be constructed for understanding a change in a particular property with the trend remaining the same. For instance, a change from $y(t) = \sin(\pi t)$ to $y(t) = 2sin(\pi t) + 1$, where the property of interest is the maximum of the trajectory $y(t_{max})$ . A top-down description would say, "The trend of the trajectory remains the same, but the maximum $y(t_{max})$ changes from 1.0 to 3.0". Note that **this may not be captured by a bottom-up approach if the maximum is not obtained at one of the pre-specified time points** (e.g., $t_{max} \notin \{0.0,0.2,0.4,0.6,0.8\}$). It also **cannot be captured if $t_{max}$ changes as well**.
> > >
> > > The information about the change in the trajectory can be read from the left part of our interface (the colorful bands or 2D contour plots), whereas the information about the properties can be read from the right part of the interface. **Please note that it does not require moving the sliders back and forth.** This information is readily available from a snapshot of the interface (as shown in Appendix E.1).
> > >
> > > These examples highlight how a top-down approach to understanding trajectory change is more intuitive and useful than bottom-up approaches.
> > >
> > > # Claims
> > >
> > > The fact that all trajectories predicted by linear regression and GAMs are parallel is a direct result of their definitions. We have already provided an explanation in Appendix E.5, and we restate it here for your convenience. Trajectories predicted by GAM-T are defined as $f(\textbf{x})(t) = f_t(t) + \sum_{m=1}^M f_m(x_m)$. The trajectories predicted by GAM-T are thus equal to the shape function $f_t(t)$ shifted vertically by $f_1(x_1) + \ldots, + f_M(x_M)$ for static features $\mathbf{x}$. Linear regression is a special type of GAM where all shape functions are linear, i.e., $f_m(x_m) = c_m x_m$.
> > > ****
> > > # Comparison with Transformer
> > >
> > > We kindly disagree with the following statement.
> > > > it is also impossible to understand how the change in the input will influence the trend and properties with TIMEVIEW.
> > >
> > > Figure 7 shows how by lowering the dose of the drug below 0.15 (**change in the input**), the trajectory changes from decreasing to increasing (**change of the trend**). Similarly, Figure 8 shows the relationship between the dose of the drug (**input**) and the minimum of the trajectory (**the property**). Note, all of this can be read **directly from the snapshots**. We do not have to move the sliders around to infer this information.
> > >
> > > We acknowledge your comment about understanding the "reason for these changes". As mentioned earlier, we interpret it as a question about simulatability, and we discuss it earlier in our response.
> > >
> > > `reply continues in the next comment...`

---

> > > > ### Author Response · Authors · 2023-11-22
> > > > **Dear Reviewer nunX [3/3]**
> > > >
> > > > `... reply continued`
> > > >
> > > > # Comparison with a linear model
> > > >
> > > > Our statement that this particular linear model would offer poor performance follows from the fact that it is a very special kind of GAM-T that we have already compared against. We suspected it would perform worse as it ignores all other features apart from the one corresponding to the initial condition. We have also now verified this empirically in the new experiments (please see below). As described in our paper and above, all trajectories predicted by GAM-T are parallel. Thus, they can only model one trend. As the focus of the synthetic datasets was on different trends, it is not surprising that GAM-T (and derivative methods) performed very poorly on them. In particular, the Tumor dataset contains trajectories of different trends (e.g., exponential increase, exponential decay, decrease and then increase). As we explained in our previous response, the linear model you refer to was proposed for more complex input data and high-dimensional trajectories. Thus, in these more complex settings, it may provide sufficient performance. **We are very clear throughout the whole paper about the exact setting our approach can be applied to (please see Section 3), which is clearly not every possible time series forecasting problem.**
> > > > ## New results
> > > >
> > > > The table below shows the part of Table 3 with the four datasets with regular measurements and an additional method (the scores refer to RMSE, lower is better). As we discussed in our previous response, the additional baseline (denoted here SimpleLinear) can only work for datasets where all trajectories are measured at the same time points. As expected, it offers very poor performance. In three out of four datasets, it performs worse than GAM-T. It performs slightly better on the Beta dataset. Note that we give the initial measurement to SimpleLinear, but we do not provide it to GAM-T, which may explain this discrepancy. As expected, we see the gap between GAM-T and SimpleLinear increases for a larger number of features (SimpleLinear ignores all additional features as it "shares weights across different variates and does not model any spatial correlations").
> > > >
> > > > | method         | flchain     | Tumor       | Sine        | Beta        |
> > > > |:-------------- |:----------- |:----------- |:----------- |:----------- |
> > > > | RNN            | 0.26 ± 0.02 | 0.02 ± 0.01 | 0.02 ± 0.01 | 0.02 ± 0.02 |
> > > > | $\Delta t$-RNN | 0.27 ± 0.01 | 0.02 ± 0.01 | 0.02 ± 0.01 | 0.02 ± 0.02 |
> > > > | XGB-T          | 0.20 ± 0.00 | 0.01 ± 0.00 | 0.53 ± 0.00 | 0.07 ± 0.00 |
> > > > | LGBM-T         | 0.20 ± 0.00 | 0.01 ± 0.00 | 0.00 ± 0.00 | 0.00 ± 0.00 |
> > > > | CatBoost-T     | 0.21 ± 0.00 | 0.00 ± 0.00 | 0.00 ± 0.00 | 0.00 ± 0.00 |
> > > > | PySR           | 0.22 ± 0.01 | 0.10 ± 0.02 | 0.05 ± 0.04 | 0.24 ± 0.03 |
> > > > | SINDy          | 0.39 ± 0.00 | 0.07 ± 0.00 | 1.30 ± 0.00 | 2.74 ± 0.00 |
> > > > | Linear-T       | 0.34 ± 0.00 | 0.68 ± 0.00 | 0.99 ± 0.00 | 1.03 ± 0.00 |
> > > > | DecisionTree-T | 0.21 ± 0.00 | 0.22 ± 0.00 | 0.10 ± 0.00 | 0.34 ± 0.00 |
> > > > | GAM-T          | 0.32 ± 0.00 | 0.54 ± 0.00 | 0.54 ± 0.00 | 0.69 ± 0.00 |
> > > > | TIMEVIEW       | 0.24 ± 0.02 | 0.00 ± 0.00 | 0.02 ± 0.00 | 0.04 ± 0.00 |
> > > > | SimpleLinear   | 0.57 ± 0.00 | 0.89 ± 0.00 | 0.78 ± 0.00 | 0.62 ± 0.00 |
> > > > ****
> > > > We hope we have addressed all your concerns. Please let us know if there is anything that requires further explanation. We are eager to address it in the very limited time remaining.
> > > > # References
> > > >
> > > > Lipton, Zachary C. "The mythos of model interpretability: In machine learning, the concept of interpretability is both important and slippery." _Queue_ 16.3 (2018): 31-57.
> > > >
> > > > Arrieta, Alejandro Barredo, et al. "Explainable Artificial Intelligence (XAI): Concepts, taxonomies, opportunities and challenges toward responsible AI." _Information fusion_ 58 (2020): 82-115.

---

### Meta-Review · Area_Chair_GPzM · 2023-12-06

**Metareview:**

This paper aims to develop a transparent model together with an interactive visualization tool for time series forecasting. The topic is important and this work has made a good attempt in the direction. Nevertheless, the presentation has room for improvement to avoid ambiguities in understanding the paper. Some claims especially in the earlier parts of the paper appear too broad and general, making people wonder whether the proposed scheme works under all circumstances. While the paper has merits that support acceptance, the authors are highly recommended to consider the comments and suggestions of the reviewers seriously in improving their paper before publication.

**Justification For Why Not Higher Score:**

It can at best be accepted for poster presentation.

**Justification For Why Not Lower Score:**

It can be rejected if there are many stronger papers.

---

### Decision · Program_Chairs · 2024-01-16

Accept (poster)